# Nearly complete structure of bacteriophage DT57C reveals architecture of head-to-tail interface and lateral tail fibers

Rafael Ayala [1,6], Andrey V. Moiseenko [2,6], Ting-Hua Chen [1,6], Eugene E. Kulikov[2,3], Alla K. Golomidova[3], Philipp S. Orekhov [4], Maya A. Street [1], Olga S. Sokolova [2,4] ✉, Andrey V. Letarov [2,3] ✉ & Matthias Wolf [1,5] ✉

The T5 family of viruses are tailed bacteriophages characterized by a long non-contractile tail. The bacteriophage DT57C is closely related to the paradigmal T5 phage, though it recognizes a different receptor (BtuB) and features highly divergent lateral tail fibers (LTF). Considerable portions of T5-like phages remain structurally uncharacterized. Here, we present the structure of DT57C determined by cryo-EM, and an atomic model of the virus, which was further explored using all-atom molecular dynamics simulations. The structure revealed a unique way of LTF attachment assisted by a dodecameric collar protein LtfC, and an unusual composition of the phage neck constructed of three protein rings. The tape measure protein (TMP) is organized within the tail tube in a three-stranded parallel α-helical coiled coil which makes direct contact with the genomic DNA. The presence of the C-terminal fragment of the TMP that remains within the tail tip suggests that the tail tip complex returns to its original state after DNA ejection. Our results provide a complete atomic structure of a T5-like phage, provide insights into the process of DNA ejection as well as a structural basis for the design of engineered phages and future mechanistic studies.

Uncontrolled use of antibiotics in medicine, animal rearing and agriculture over more than half a century has contributed to the emergence and world-wide spread of multidrug-resistant strains of pathogenic bacteria that threaten the efficacy of current antibacterial therapies[1]. Therefore, the development of alternative technologies which can replace or complement antibiotics is paramount. One promising strategy is the biological control of pathogen populations using bacteriophage-based preparations[2,3]. To enable phage therapy by

design, directed modification of well-characterized phage strains (phage platforms) active against groups of microorganisms of interest has been attempted by methods of genetic engineering and synthetic biology[4]. A key aspect of this strategy is the need for comprehensive study of bacteriophages including structure-guided phage engineering[5].

Most bacteriophages harnessed to combat infections so far belong to the group of tailed phages. Tailed bacteriophages are

[1]Molecular Cryo-Electron Microscopy Unit, Okinawa Institute of Science and Technology Graduate University, 1919-1 Tancha, 904-0495 Onna-son, Okinawa, Japan. [2]Faculty of Biology, Lomonosov Moscow State University, 1 Leninskie Gory, Bld. 12, 119234 Moscow, Russia. [3]Winogradsky Institute of Microbiology, Research Center of Biotechnology of the Russian Academy of Sciences, 7/2, 60-Letiya Oktyabrya Ave, 117312 Moscow, Russia. [4]Faculty of Biology, Shenzhen MSU-BIT University, 1 International University Park Dr, Dayun New Town, Longgang District, Shenzhen 518172, China. [5]Institute of Biological Chemistry, Academia Sinica, 128 Academia Road Sec. 2, 115, Taipei 15, Taiwan. [6]These authors contributed equally: Rafael Ayala, Andrey V. Moiseenko, Ting-Hua Chen. ✉e-mail: sokolova@mail.bio.msu.ru; letarov@gmail.com; matthias.wolf@oist.jp

classified *Caudoviricetes*[6], representing the most prolific virus group in the biosphere and, more generally, the most abundant and diverse life form on Earth[7,8]. The hallmark of *Caudoviricetes* is the presence of a tail, responsible for attachment to the host cell and for paving the way for injecting the viral genomic DNA through multiple layers of bacterial cell wall. The mode of action of the molecular machinery of the tail has been revealed for the contractile systems[9,10], but it is much less understood in podoviruses and siphoviruses, which do not use a contractile tail. Siphovirus T5 is a virulent phage that infects *Escherichia coli* and was included in the archetypical T-series of model *E. coli* phages[11]. While the structure of most of the tail tip of T5 was recently solved[12], it did not resolve the detailed structure of its lateral tail fibers (LTFs). Furthermore, the architecture of the connection between the tail and the capsid remains unknown.

Bacteriophage DT57C is closely related to phage T5, with most of its structural proteins having direct analogs in both viruses (Table 1). However, DT57C has been shown to possess a different allelic type of the main receptor-binding protein Hrs (pb5), recognizing BtuB instead of the FhuA receptor. This phage infects several *E. coli* strains with complex O antigens of the O22 and O81 types[13]. The genetic and mutagenesis data indicate that the LTFs of this virus have a branched organization comprising two proteins, LtfA and LtfB, with LtfA connected to the phage tail while LtfB is attached via LtfA (in contrast to the single Ltf protein in the phage T5). These LTFs are essential for phage viability in conjunction with O antigen-producing strains, but not for the rough host strains such as *E. coli* C600[13,14]. Interestingly, although LtfA and LtfB proteins are very distant from T5 Ltf by their amino acid sequences, the N-terminal moiety of LtfA is closely related to the N-terminus of the phage T5 Ltf protein. DT57C phage LtfC protein, which is essential for LTF attachment, is identical to its T5 homolog. These observations indicate that the joint between the LTFs and the body of the tail is conserved between DT57C and T5 (and among T5-related phages in general). The bulk of the virion core (capsid, tail shaft and tail tip) proteins of DT57C are closely related (sequence identity 85-100%), and therefore the structural insights for these components may be extended to the paradigmal T5 bacteriophage.

Here, we aimed to determine the molecular architecture of the different regions of the T5-like phage DT57C, and to obtain additional functional insights by further examining the structures through all-atom molecular dynamics simulations, which have been recently revealed insights into the mechanisms through which biological macromolecules perform their functions[15–20]. We present a composite map determined by electron cryo-microscopy (cryo-EM) at resolutions ranging between 2.9 Å to 4.9 Å of the entire T5-like phage particle DT57C, and atomic models comprising its core structural proteins. Our reconstructions revealed an atypical way of attachment of the lateral tail fibers to the tail tip, and the structure of the head-to-tail interface (neck) assembled in three protein rings, instead of four as observed for several other phages[21,22]. Furthermore, we could separate the particle population into pre-ejection (capsid filled with DNA) and post-ejection (empty capsid after DNA release) states and to reconstruct the related states of the portal complex and the tail tip individually at resolutions sufficient to build atomic models of each state, gaining insight into the conformational changes leading to DNA release.

## Results

### Capsid

A combination of strategies aimed at reconstructing different parts of the virus by single particle cryo-EM resulted in a nearly complete atomic model of the DT57C virion (Fig. 1). Facilitated by the icosahedral symmetry of the capsid, it was reconstructed at 2.9 Å resolution. Although one of the vertices of the capsid is occupied by the portal complex, icosahedral symmetry was applied to maximize resolution. Consistent with structures of the closely related T5 phage[23], the $T = 13$

icosahedral capsid of mature DT57C was composed of two proteins, the major capsid protein (MCP) and the head decorating protein (DCP), and it had a diameter of 930 nm (Supplementary Fig. 1a). Our high-resolution map led to an atomic model with improved geometry. The asymmetric unit comprises 13 copies of the MCP lacking the Δ-domain (the first 160 N-terminal residues, which is cleaved during the maturation process of the capsid). While one of them is located at an icosahedral vertex and contributes to the formation of pentamers, the remaining 12 form two consecutive proximal and distal hexamers (Supplementary Fig. 1b).

We observed clear additional map features located at the center of each hexamer, corresponding to the decoration protein. The DCP of DT57C is closely related to the pb10 protein of T5, with the main difference being the insertion of an additional, divergent Ig-like domain between the N-terminal capsid-binding domain and the C-terminal Ig-like domain. The strongest part of the electron potential map was found at the binding pocket in the center of each hexamer, corresponding to the N-terminal domain. Ig-like domains can emerge at lower contour level (Supplementary Fig. 2). Interestingly, while the N-terminal domain was not resolved well enough to build an atomic model, it displayed local pseudo-2-fold symmetry (Supplementary Fig. 2). This suggests that the DCP may bind in preferred orientations. To further investigate the binding mode of the DCP to the MCP hexamers, we calculated the Coulombic electrostatic potential of the two MCP hexamers of the asymmetric unit (Supplementary Fig. 3). The center of both hexamers, located below the map feature corresponding to the N-terminal domain of DCP, was predicted to be highly negatively charged. Indeed, the sequence of the N-terminal domain of the DCP contains multiple positively charged residues, forming a cluster at positions 50-55 (RKVWKK). Based on the highly negative electrostatic potential at the center of both hexamers, it is likely that the DCP binds to it through such positively charged regions.

### Capsid-tail interface

The reconstruction of the capsid-tail interface, here referred to as the neck region, reached a resolution of 3.4 Å and revealed the presence of three proteins that enable the transition from capsid to the tail tube. The portal protein (PrtP) forms a dodecameric ring, which replaces one of the MCP pentamers of the capsid. The PrtP ring interacts directly with a ring of head completion protein (HCP), also dodecameric. The interface is completed by a hexameric ring of tail completion protein (TCP), which connects to the first trimeric ring of tail tube protein (TTP) (Fig. 2). Additionally, we observed a strong density emerging from the interior of the capsid through the portal lumen, corresponding to DNA (Fig. 2).

The PrtP is organized in three structural domains (Supplementary Fig. 4a). Domain DI is the largest central domain, which mediates capsid attachment. However, symmetry mismatches between the capsid and the neck prevented us from obtaining a well-resolved map within the region of direct interaction between PrtP and the surrounding MCP hexamers. The domain DII interacts with the HCP, and DIII creates the capsid-proximal part of the inner pore through which the DNA exits from the capsid (Fig. 2).

The HCP comprises two domains (Supplementary Fig. 4b), DI and DII. DI consists mostly of alpha helices and participates in one of the interfaces between PrtP and HCP. It also contains a β-hairpin (138-149) facing the inner lumen of the tube. DII is a small β-barrel, which mediates another interaction area with the PrtP. Two specific interface areas are therefore established between PrtP and HCP (Supplementary Fig. 5). The first one involves an α-helix from PrtP DII (216-230), which interacts with the HCP small β-barrel. The second one consists of a loop from PrtP DII (215-211) contacting one of the α-helices from HCP DI (158-167).

Finally, the TCP is structured as a single compact domain (Supplementary Fig. 4c), with a β-sheet facing the lumen of the tube. The

**Table 1 | Comparison of structural proteins in bacteriophage DT57C and bacteriophage T5 systems**

| Function (synonym) | Functional name | Protein band (pb) | DT57C gene product (NC_027356) | T5 homolog (NC_005859) | Degree of DT57C-T5 identity/similarity | Matching regions |
|---|---|---|---|---|---|---|
| Main capsid protein | MCP | pb8 | gp121 | gp149 | 95% / 97% | H97-like MCP fold (61-442) |
| Head decorating protein | DCP | pb10 | gp123 | gp151 | 62% / 63% | Hexon-binding (1–64); Ig-like (65-142) |
| Portal protein | PrtP | pb7 | gp124 | gp152 | 98% / 99% | Wing (1-175, 289-326); Stem (176-199, 256-288); Clip (200-255); Crown (327-378) |
| Head completion protein | HCP | - | gp120 | gp148 | 97% / 99% | DI (1-43, 112-170); DII (44-111) |
| Tail completion protein | TCP | - | gp119 | gp147 | 93% / 98% | DI (2-155) |
| Tail tube protein(main tail protein) | TTP | pb6 | gp117 | gp145 | 85% / 92% | DI (3-373), Ig-like (376-468) |
| Tape measure protein | TMP | pb2 | gp113 | gp140 | 67% / 78% | C-terminal coiled-coil (1192-1227) |
| Tail tip middle protein (tip middle protein) | TTMP | - | gp116 | gp144 | 72% / 85% | DI (1-71, 100-125, 134-164); DII (85-99, 126-133, 165-197) |
| Distal tail protein | Dit | pb9 | gp112 | gp139 | 82% / 91% | DI (85-99, 126-133, 165-297); DII (334-378) |
| Tail tip hub protein | TTHP | pb3 | gp111 | gp138 | 84% / 93% | DI (1-163, 550-590, 658-711); DII (164-549, 591-657, 712-736); Ig-like DIII (737-949) |
| Central tail fiber protein (straight fiber) | CFP | pb4 | gp110 | gp137 | 77% / 87% | Ig-like DI (1-326); DII (327-461); DIII (462-685) |
| Receptor binding protein | RBP | pb5 | gp128 | gp157 | 31% / 43% | - |
| Lateral tail fiber protein A | LtfA | (pb1 in T5 only) | gp108, LtfA | gp133, Ltf[a] | 22% / 35% | - |
| Lateral tail fiber protein B | LtfB | - | gp107 | – | – | - |
| Lateral tail fiber assembly protein | LtfC | - | gp109 | gp135, 136[b] | 75% / 88% | DI (1-140) |

[a]Only the N-terminal fragment of the LtfA protein in phage DT57C is homologous to the Ltf of phage T5.
[b]Our sequence of DT57C suggests a frame shift error in the homolog reading frame of the LtfC T5 genome sequence (GenBank RefID 277764).

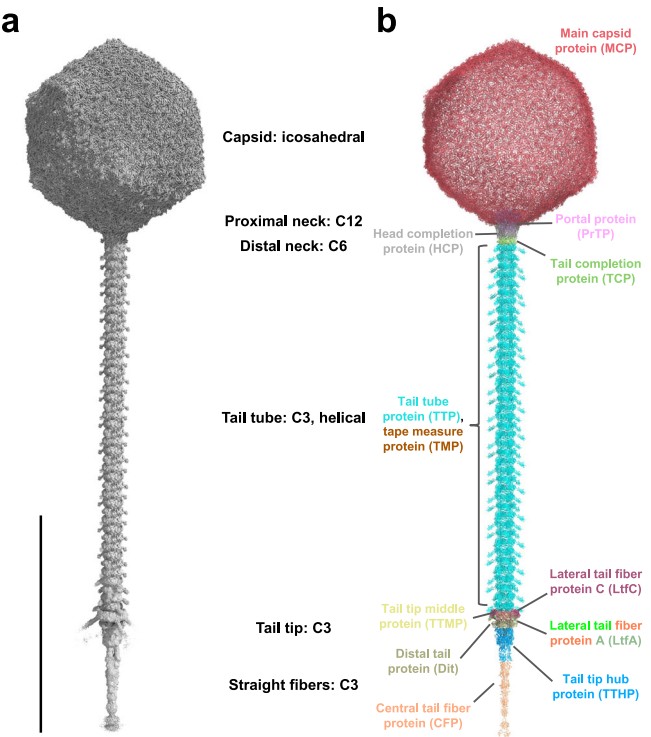

**Fig. 1 | Architecture of the DT57C bacteriophage. a** Iso-electron potential surface of a composite cryo-EM map of the entire virion. Six individually symmetrized focused maps (containing capsid, portal, distal neck, tail, tail tip and straight fiber regions) were reconstructed at resolutions between 2.9–4.9 Å and combined into a composite map (see Supplementary Information and Methods). The normalized (average density 0, standard deviation 1) composite map was contoured at 3.0σ above average. **b** Refined atomic model of the entire phage DT57C in ribbon representation. A total of 12 gene products are distinguished by colors which are consistently used throughout the manuscript. The scale bar represents a length of 1000 Å.

interaction between the TCP hexameric ring and the HCP dodecameric ring is mediated by DI of two adjacent HCP monomers, which contact the top part of a single TCP monomer (Supplementary Fig. 6). Each TCP monomer inserts wedge-like into the surrounding cleft formed by the DI from HCP monomers, with the interaction surface involving a large number of charged residues from both partners.

The TCP hexameric ring therefore acts as an adapter to transition between the neck area with C12 symmetry (composed by PrtP and HCP) and the tail tube with C3 symmetry. Even though the resolution of the neck reconstruction at the TCP-TTP interface was not high enough to allow de novo model building of the first TTP ring, we could dock an atomic model of the TTP ring built based on our helical tail tube reconstruction, thus identifying the interaction interfaces between TCP and the first TTP ring (Supplementary Fig. 7).

At the HCP-TCP interface level, there is a steep reduction in the diameter of the density attributed to the DNA (Fig. 2b), suggesting that DNA is locked in the PrtP-HCP lumen through interactions with the DI of the HCP ring even before attachment of the tail to packed viral heads. Indeed, the DI of HCP possesses flexible β-hairpins which could potentially flip inside the pore, partially occluding it. To check this hypothesis, we performed a molecular dynamics simulation of the HCP ring alone, which revealed that the β-hairpins of the DI of HCP are prone to large-scale fluctuations resulting in conformations partially occluding the central pore (Fig. 2d, Supplementary Movie 1, Supplementary Fig. 8). This is in accordance with the fact that packed heads must be in a closed state to prevent ejection of the DNA[24]. Additionally, the electrostatic potential inside the channel of the portal complex

(Fig. 2e) shows negative values along the entire inner space of the channel. The negative potential reaches a minimum at the level of the HCP-TCP interface. Since the negatively charged DNA should experience significant repulsive force when passing through this region of the portal complex, the dynamic nature of the HCP DI β-hairpins may contribute towards preventing premature DNA release.

The β-hairpins of HCP did not undergo any significant fluctuations in the simulation containing TCP in complex with HCP, indicating that the former can hinder pore closure as required to establish a continuous channel from head to tail tip upon insertion of the tail into packed heads (Fig. 2d). A detailed examination of both HCP simulations revealed that both observed conformations for the β-hairpins of HCP are stabilized through ionic bridges between highly conserved residues of HCP (K140 with D135 or E145, respectively for the open and closed pore conformations) (Supplementary Figs. 9 and 10, Supplementary Movie 2). Moreover, the presence of the TCP ring further stabilizes the open conformation due to formation of an alternative ionic contact between E145 of HCP and K41 of TCP (Supplementary Fig. 11).

The DNA density is immediately followed by additional density, which was assigned to the TMP N-terminus (colored brown in Fig. 2b, Supplementary Fig. 12a). While the exact interface between the TMP N-termini and the DNA could not be fully resolved, the contact between them was clearly apparent. Furthermore, our cryo-EM map resolved the secondary structure of the N-terminal part of the TMP as a trimeric α-helical coiled-coil.

## Tail tube

The tail of DT57C is composed of 40 stacked rings of tail tube protein (TTP) (Supplementary Fig. 13), with each ring containing three subunits. In addition to analysis of individual particles by cryo-EM, the number of rings composing the tail was verified by electron cryotomography (Supplementary Movie 3). The tail has helical symmetry and 3-fold rotational symmetry, with a rise 41.8 Å and twist 39.5°. The TTP shares 85% amino acid sequence identity with the T5 pb6 protein[21], which is composed of a tail tube domain and the protruding Ig-like decoration domain (Fig. 3). While the local resolution of our reconstruction at the flexible Ig-like domains was too low for de novo model building, a predicted structure fitted the molecular envelope well. The tail shaft structure of DT57C is nearly identical to a previously published T5 tail tube structure[21].

To better address tail flexibility and curvature of the segments, we performed asymmetrical reconstruction without imposition of any symmetry (Fig. 3d), which clearly revealed the presence of TMP inside the tail.

## Tail tip complex

The tail tip complex has overall C3 symmetry, and its core is composed of five different proteins organized in three layers: lateral tail fiber protein C (LtfC), lateral tail fiber protein A (LtfA), tail tip middle protein (TTMP), distal tail protein (Dit) and tail tip hub protein (TTHP) (Fig. 4).

The first layer directly interacts with the last TTP ring and is composed of an inner trimeric ring of TTMP surrounded by a dodecameric ring of LtfC (Fig. 4). TTMP has a compact structure composed mostly of β-sandwiches arranged in two subdomains separated by an α-helix (Supplementary Fig. 14a). It bears considerable similarity to the fold of the TTP, differing mostly in the loops and helices linking the core β-sandwiches, which suggests a potential common evolutionary origin for the two proteins. Interactions between the last TTP ring and TTMP are established through two long protruding β-hairpins of the of TTP which contact the top of TTMP monomers (Supplementary Fig. 15).

The inner trimeric TTMP ring is surrounded by an outer ring composed of 12 LtfC monomers. LtfC adopts a compact, single-domain structure composed mostly of β-sheets (Supplementary Fig. 14b), with

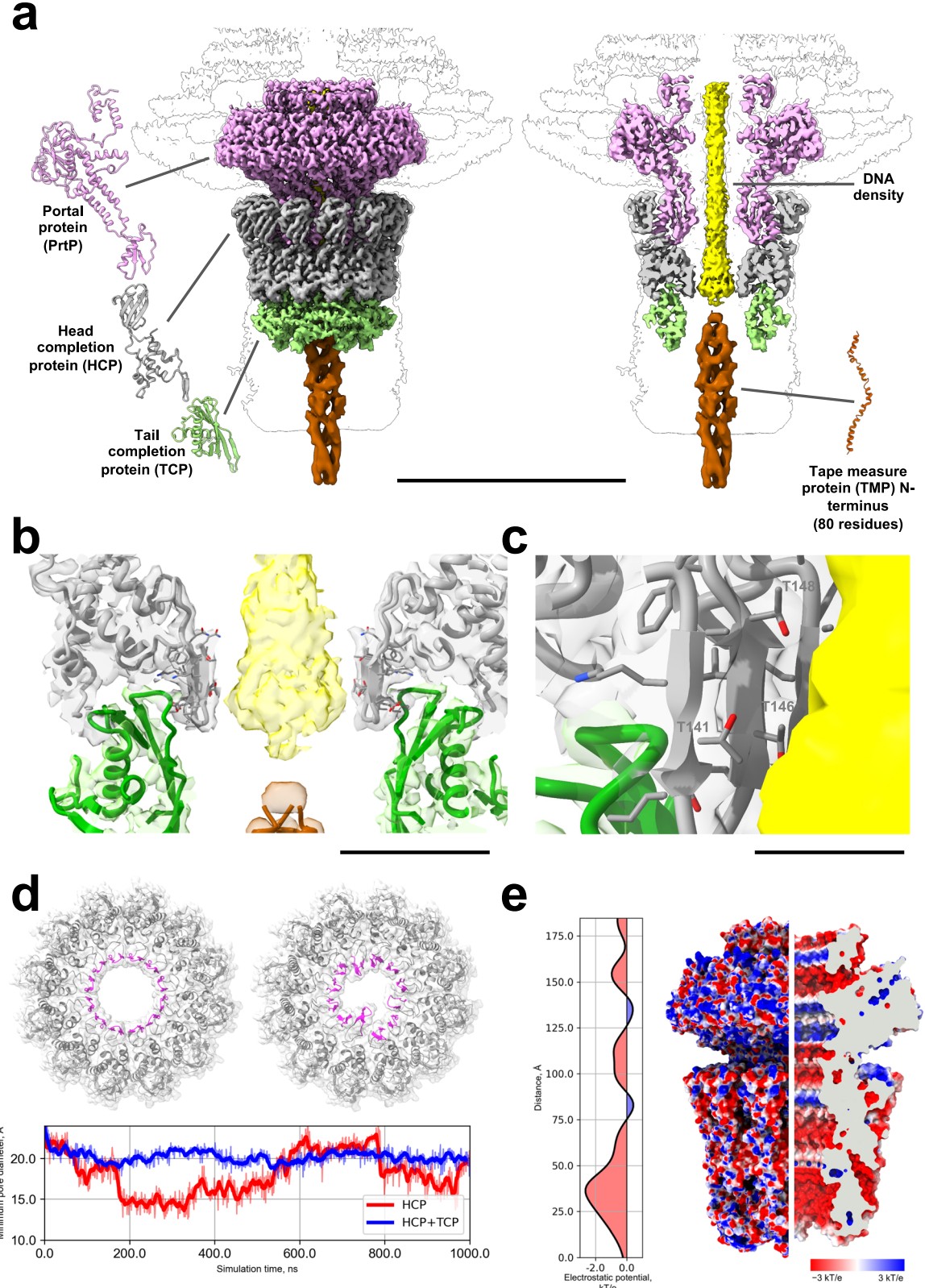

each TTMP monomer contacted by four copies of LtfC (Supplementary Fig. 16). A cavity remains between the TTMP and the part of the LtfC ring distal to the tail tube, into which LtfA is inserted. LtfA is characterized by the presence of a long α-helix, which starts at residue T40 and acts as a trimerization domain. The trimeric bundle of α-helices, contributed by three different copies of LtfA, protrude radially outwards from the tail tip complex (Fig. 5, Supplementary Fig. 14c).

The N-terminal region of LtfA preceding the α-helical part adopts a structure characterized by two β-strands connected by flexible linkers. The specific arrangement of the β-strands and its connectors varies between the three different LtfA polypeptide chains contributing to each LtfA trimer (Supplementary Fig. 14c). The N-terminal portions of LtfA form a lasso-like structure filling the groove between the LtfC and TTMP rings, such that each LtfA copy extends in opposite

**Fig. 2 | Structure of the DT57C bacteriophage head-to-tail interface. a** Overview of the head-to-tail interface (neck). The neck comprises three proteins: PrtP (pink), HCP (gray) and TCP (green). PrtP is directly embedded in the capsid and forms a dodecameric ring which is inserted onto a second dodecameric ring formed by HCP. The TCP hexameric ring acts as an adapter between the dodecameric symmetry of the HCP layer and the trimeric symmetry of the tail tube. DNA density is observed until the end of the HCP layer, where it is immediately followed by the N-terminal part of the TMP. **b** Area of transition between DNA and TMP N-terminus at the level of the interface between HCP and TCP. The transition (observed in unmasked maps) can be clearly observed as a sudden change in the diameter of the central density. **c** Residues of HCP in close proximity to DNA include multiple threonines, which are known to interact frequently with the DNA backbone. **d** Top, cross-section of the HCP ring in the open (left) and occluded (right) states during the molecular dynamics simulation of HCP only, with β-hairpins facing the portal lumen. Bottom, evolution of the minimum pore diameter during the molecular dynamics simulations of HCP only (red) and HCP + TCP (blue). **e** Electrostatic potential (ESP) profile inside the portal complex (left) and ESP mapped on the outer and inner surfaces of the portal complex (right). Clipping plane, gray. The scale bars represent lengths of 150 Å (**a**) 30 Å (**b**) and 10 Å (**c**). Source data for (**d**, **e**) are provided as a Source Data file.

direction as observed from the outer end of the helical bundle (Fig. 5). The capsid-distal LtfA contacts the capsid-proximal LtfA from an adjacent LtfA trimer, enabling the formation of a closed ring. The β-strands of all LtfA copies interact with the LtfC ring by augmenting their β-sheets of the latter to eight- or nine-stranded sheets by donating one or two strands from LtfA (Fig. 5, Supplementary Fig. 17).

To study the overall stability of this complex, we performed an equilibrium molecular dynamics (MD) simulation of the TTMP tail protein ring along with the adjacent LtfA-LtfC ring. The simulation was run for 1 μs and it revealed that the LtfA/LtfC/TTMP backbone deviations did not exceed 5 Å (mean value 4.3 Å over the last 500 ns) throughout the simulation (Supplementary Fig. 8). Nonetheless, we detected partial unfastening of one LtfA trimer involving two LtfA monomers out of three, with their proximal LtfC extending outwards from the central TTMP moiety by up to 20 Å (Supplementary Fig. 18, Supplementary Movie 4). Such conformational change could be involved in the mechanism for opening a channel for the ejection of DNA upon host recognition. The results of these simulations provide indirect evidence that the regions of LtfA, which appear in the initial model as extended, largely preserve their secondary structure. Moreover, these regions tend to expand when neighboring residues are also adopting the β-sheet conformation (Fig. 5d, Supplementary Fig. 19). Remarkably, the initially unstructured N-terminal fragments of LtfA monomers turned into β-sheets in the simulation, which allowed them to form a strand-to-strand contact between the neighboring trimers, apparently stabilizing the entire LtfA ring.

Although only the N-terminal moiety of the LtfA protein is present in our reconstruction, no other robust feature of the electron potential map that could correspond to the LtfB protein was found in association with the core part of the virion. This supports the previous conclusion that the LtfB protein is attached via LtfA forming a branched LTF that markedly distinguishes DT57C from T5[13]. However, when the tail tip map was rendered at a low contour level (Supplementary Fig. 20) we observed weak densities corresponding to three fibers attached to the tail tip, potentially corresponding to LtfB. This arrangement is compatible with a complex LTF morphology of the DT57C phage studied previously[13].

The loops of the two long β-hairpins of TTMP extend tailward and contact the next ring, composed of six copies of Dit. The interaction, reminiscent of that established between TTP and TTMP, is such that each long TTMP loop contacts a Dit monomer, enabling the transition between the trimeric TTMP ring and the hexameric Dit ring (Supplementary Fig. 21).

Finally, a trimer of TTHP closes the tail tip complex (Fig. 4). It is the largest protein of the tail tip, comprising three main domains (Supplementary Fig. 14e). The capsid-proximal domain, DI, shares structural similarity with the TTP and TTMP. The Dit contacts DI of TTMP through the stacking of two β-hairpins onto the β-sandwiches of TTMP (Supplementary Fig. 22). DII of TTMP encloses a cavity where additional density was found, which we assigned to the C-terminus of the TMP. Finally, DIII comprises two fibronectin type III subdomains, which serve as connection sites for the central tail fiber protein (CFP).

To obtain a reconstruction of the CFP region, we re-extracted particles after shifting the center of the reconstruction towards the capsid-distal region of the tail tip complex and performed a refinement with local angular searches. Even though the resulting reconstruction did not allow de novo building of the CFP protein due to the high flexibility of the region, it accommodated well a predicted structural model (see Methods section). The CFP trimer is connected to the TTHP with its own fibronectin domains. Being separated at the proximal part, the CFP monomers merge together as they extend away from the core of the tail tip complex, forming a straight fiber with high β-strand content in a 3-fold symmetric arrangement (Supplementary Fig. 23).

The cavity enclosed by the TTHP also contained additional density for a small α-helical coiled coil region (Fig. 4, Supplementary Fig. 12b), which was assigned as the C-terminal part of the TMP based on its location within the lumen of the tail tube tip, the fact that this region is predicted α-helical, and a close fit to the AlphaFold model.

## Comparison of pre- and post-ejection states

Using 2D classification, we identified a smaller population of phages (~20% of all particles), where their DNA had been ejected. Following a similar methodology as for the intact phages (representing the pre-ejection state), we obtained reconstructions of the different parts of these empty phages (corresponding to the post-ejection state). We then compared the reconstructions of both states to gain insights into the changes triggered by the ejection of the genetic material. The neck reconstructions showed no significant differences between the two states other than the expected absence of DNA and TMP from the lumen of the neck and tail tube (Fig. 6a). The helical parameters of the tail tubes from both states were identical, in accordance with previous observations on the T5 phage[25].

The reconstructions of the tail tip complex did not reveal any significant difference in the conformation of their proteins either. However, interestingly, we found the C-terminal region of the TMP to remain enclosed in the tail tip surrounded by TTHP after TMP and DNA ejection (Fig. 6b). This suggests that the C-terminal part of TMP is proteolytically cleaved during the assembly of the viral particles. Proteolytic processing of TMP during phage assembly has indeed been demonstrated at one of the cleavage sites in proximity to its C-terminus[26]. As described previously, this retained C-terminal TMP fragment adopts a clearly resolved triple-stranded parallel α-helical coiled coil motif within the channel of the TTHP (Fig. 6b, Supplementary Fig. 12b).

To further confirm that empty phages indeed correspond to a post-ejection state where both the DNA and the bulk of the TMP had been properly ejected through the tail, we compared tomograms of virions in three different states: empty virions with an intact capsid, filled virions with an intact capsid and empty virions with a broken capsid (Supplementary Fig. 24). The tomograms revealed that nearly all empty virions have an intact capsid. Furthermore, empty virions with intact capsids did not contain any TMP density inside the tail tube lumen. Filled capsids, on the other hand, displayed a filled tube with density corresponding to the TMP. Finally, while a virion with a broken capsid did not contain DNA due to it being released through the breakage point of the capsid, it displayed a solid tail tube similar to that observed in filled virions, indicating that the TMP was still present in

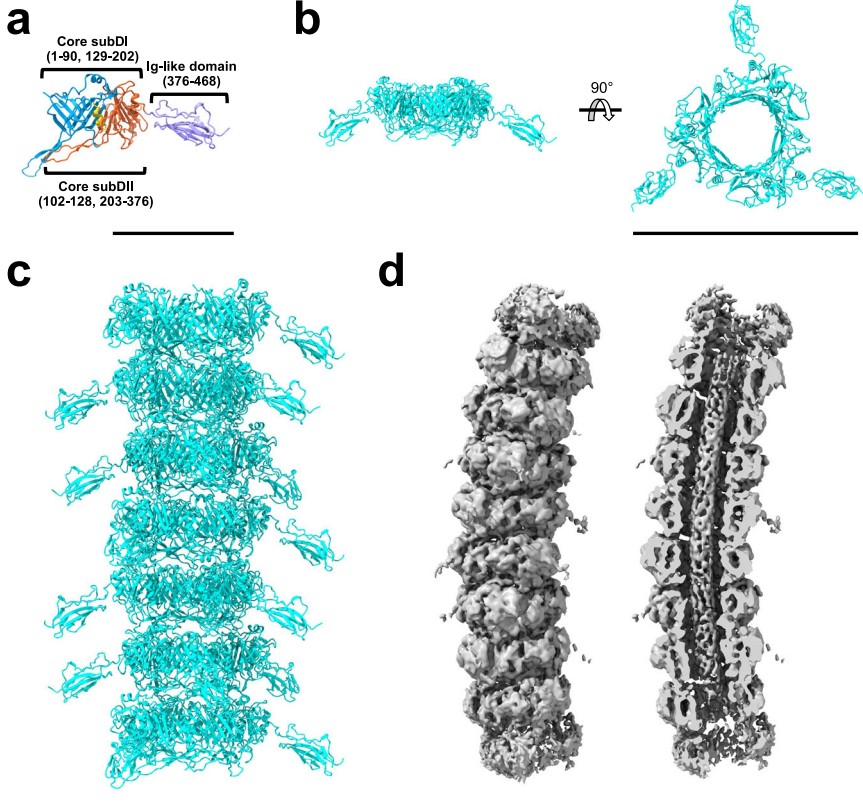

Fig. 3 | **Structure of the tail tube. a** Domain architecture of a TTP monomer.
**b** Trimeric ring of TTP. **c** Tail tube segment with seven trimeric rings of TTP.
Trimeric rings stack on top of each other, forming a helical tube with rise 41.84 Å
and twist −39.51°. **d** Asymmetric reconstruction of the tail tube. The asymmetric
reconstruction clearly revealed the presence of the TMP inside the tail tube. We
could not detect any obvious interactions between the inner wall of the tail tube
and the TMP in this asymmetric reconstruction of 40 nm long curved segments,
suggesting that the TMP may be held in place through non-periodic interactions
with the inner walls of the tail tube in addition to contacts established at its C- and
N-terminal sections. The scale bars represent lengths of 60 Å (**a**) 150 Å (**b**, **c**) and
100 Å (**d**).

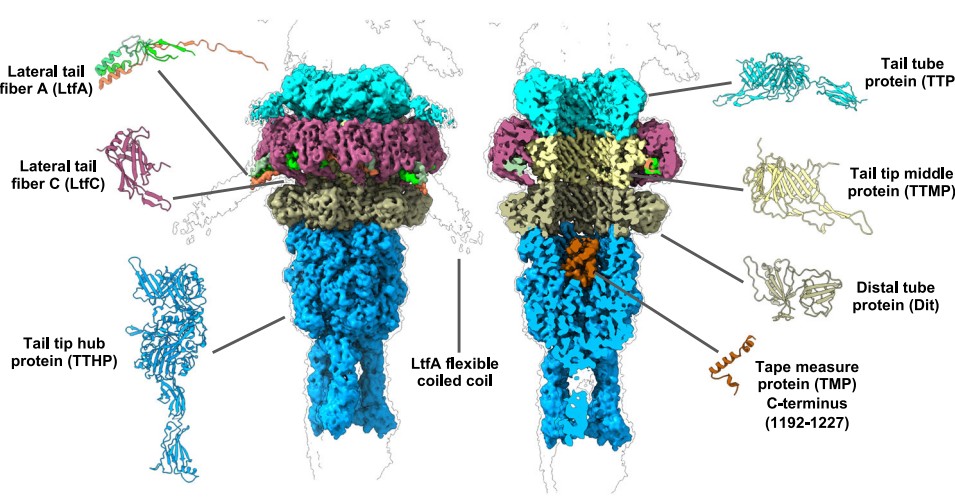

Fig. 4 | **Structure of the DT57C bacteriophage tail tip.** The first layer of the tail tip
complex comprises a trimeric ring of TTMP (light yellow), which directly interacts
with the last TTP ring (cyan) and is enclosed by an outer LtfC dodecameric ring
(maroon). A lasso-like ring composed by nine copies of LtfA (bright green, dim
green and orange) is enclosed between the TTMP and LtfC rings. The TTMP ring is
stacked on top of a hexameric Dit ring (dark yellow), which is followed by a trimer
of TTHP (blue). TTHP seals the tail tip, and contains the C-terminal portion of TMP
(brown), which is resolved as a coiled-coil helical structure. The scale bar represents
a length of 150 Å.

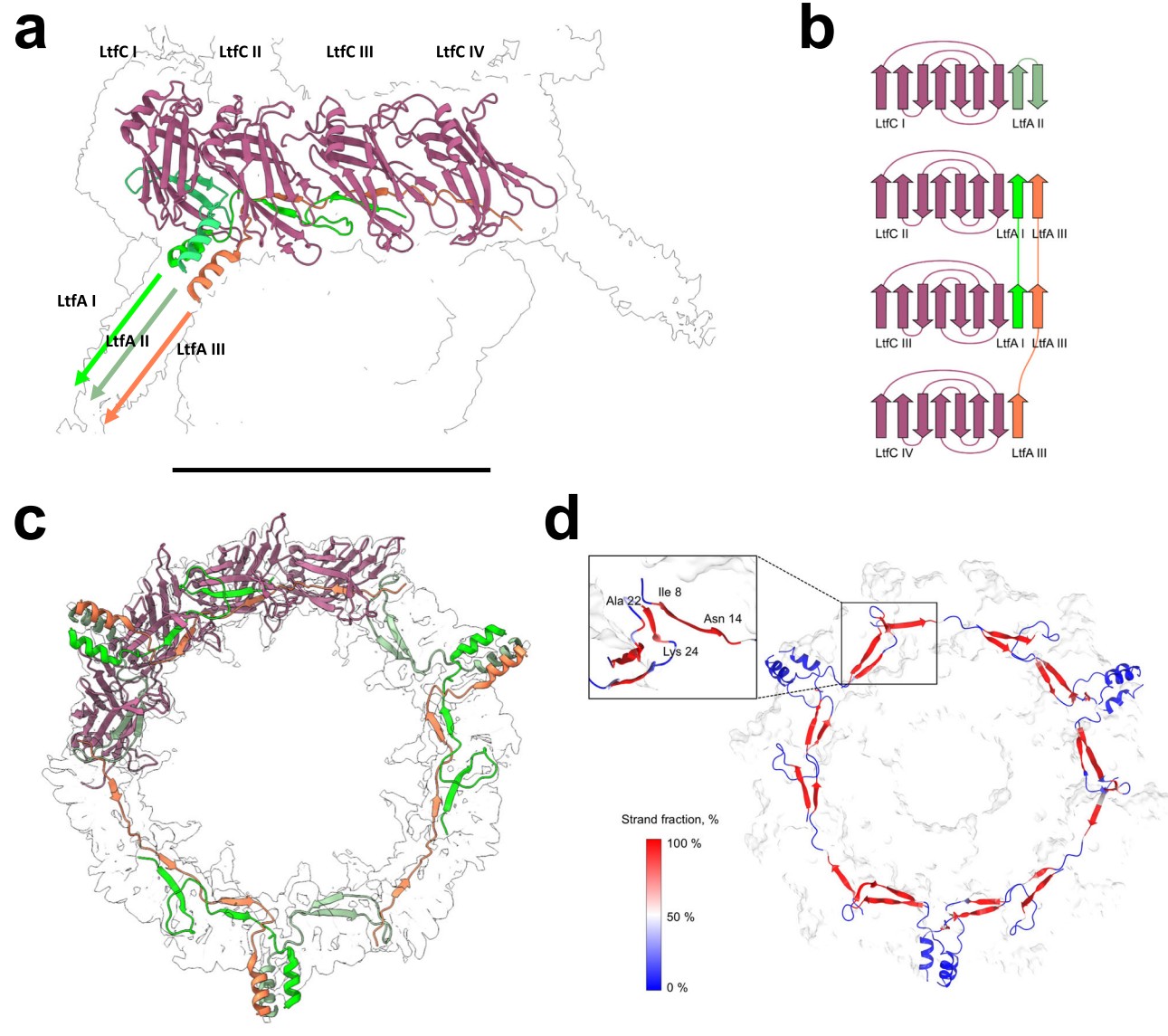

**Fig. 5 | Long tail-fiber connection site.** The tail tip reconstruction shows the detailed structure and interaction interfaces of two of the long tail-fiber (Ltf) proteins, LtfC and LtfA. **a** The N-terminal domains of each LtfA trimer are placed in the pocket enclosed by four LtfC subunits, with the rest of the trimer pointing radially outwards from the tail tip, and forming a three-helix coiled coil. **b** Each of the LtfA monomers is connected to the LtfC assembly in a unique way by augmenting the edges of LtfC β-sheets with two short β-strands. **c** The N-termini of all three LtfA trimers intertwine and form a lasso-like ring located at the cleft between the LtfC and TTMP rings. **d** Molecular dynamics simulation indicates that LtfA-LtfC β-sheets are stable over the simulation time of 1 μs and can form inter-chain strand-to-strand contacts. The scale bars represent lengths of 150 Å (**a**) 100 Å (**b**) and 110 Å (**c**).

the tail tube in spite of DNA being released through alternative channels.

## Discussion

Cryo-EM has enabled high-resolution structural analysis of viruses at near-native conditions. However, many of the studies performed so far have been limited to specific well-ordered parts, such as the icosahedral capsid or straight helical tails, missing regions especially in the case of viruses that contain large, flexible components or parts with multiple levels of symmetry[9,27–29]. In the present work, we have applied a set of focalized strategies to reconstruct the different regions of complete viral particles while overcoming the challenges imposed by symmetry mismatches and the high tail flexibility of siphoviruses. Assembling the resulting reconstructions into a composite map enabled of the complete virion of the T5-like bacteriophage DT57C by single particle cryo-EM while separating their biological states (Fig. 1).

A unique feature of DT57C is the mode of attachment of the lateral tail fibers to its tail tip. To our knowledge, in all currently available phage structures, the components of the receptor-recognition devices are attached to their baseplates in myoviruses and siphoviruses, or to the head-to-tail interface proteins in podoviruses, via joint-like interfaces formed by globular protein domains. For example, so are shaped the interface of the long and short tail fibers in T4[9,30], the fibers of the Listeria phage A511[28], *Lactococcus* siphoviruses TP901-1[31] and p2[32], *Escherichia* podoviruses P22[33] and T7[34] and in many other bacteriophages.

In DT57C, we found that LTF attachment is mediated by the dodecameric LtfC ring surrounding the TTMP (Figs. 4 and 5). The groove formed between the LtfC and TTMP subunits accommodates the polypeptide chains of the LtfA protein that donate their β-strands to the β-sheets of the LtfC subunits (Fig. 5b). The LtfA N-terminal moieties form a lasso-like structure within the groove formed by the

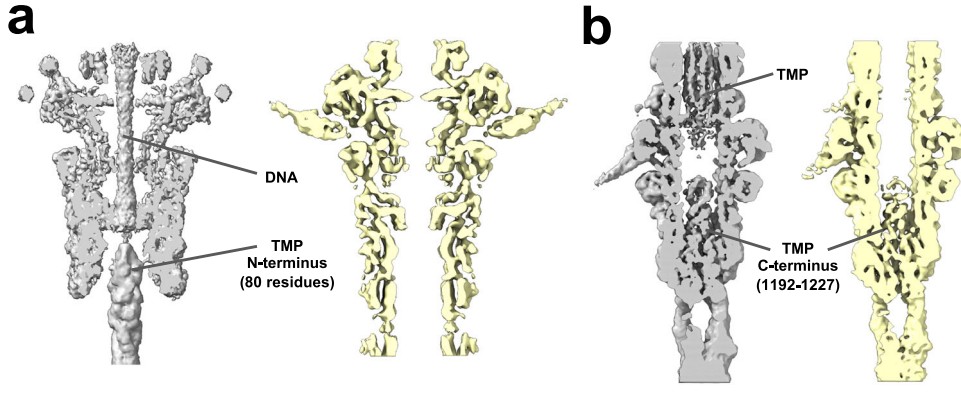

**Fig. 6 | Comparison of the pre- and post-ejection states.** Reconstructions of the pre-ejection state are shown in gray, while those of the post-ejection state are shown in yellow. **a** Neck (portal interface) density map. Densities for the tape measure protein (TMP) N-terminus and DNA are absent in the reconstruction of post-ejection phages. **b** Tail tip density map. The TMP C-terminus density is present in both states, but the density in the channel is not visible in phages after ejection. The scale bars represent lengths of 150 Å (**a**) and 100 Å (**b**).

LtfC subunits (Fig. 5c). Such deep entanglement of 12 LtfC chains and nine LtfA chains into the common structure would make LTF attachment very robust and exclude the possibility of an incomplete LTF number. Three N-terminal fragments from each of the LtfA chains that are identical by their amino acid sequence adopt different conformations upon entering the LtfC ring. Another example of different conformation adopted by identical polypeptide chains was previously reported for the baseplate protein gp10 in bacteriophage T4[9].

Although only the N-terminal moiety of the LtfA protein is present in our reconstruction, no other robust electron density that could correspond to the LtfB protein was found in association with the core part of the virion. This finding indirectly confirms the previously drawn conclusion that the LtfB protein was attached via LtfA, forming a branched LTF that markedly distinguishes DT57C from T5[13]. However, when the tail tip map was rendered at a low contour level (Supplementary Fig. 20), we observed weak protrusions corresponding to three fibers attached to the tail tip. We interpret these map features as LtfB trimers that may be folded back and interacting with the tail tip by their C-terminal ends. This arrangement is compatible with a complex LTF morphology of the DT57C phage studied previously[13]. The biological significance of such an interaction is not clear. However, the folded back LtfB should not be able to interact with the receptor (O81 type O antigen)[13]. If the interaction of LtfB with the tail tip were modulated by any environmental cues such as chemical signals, pH value, ionic strength or temperature, a simple environmental sensor may be present in the DT57C virion to control phage infectivity on the host strains recognized by LtfB. Similar environmental sensors based on the interaction of the lateral tail fibers with the core part of the tail and whiskers have been described in T4 and some related bacteriophages[35].

To gain insights into the mechanism of DNA ejection in DT57C, we compared the reconstructions of each region from full and empty viruses, representing pre- and post-ejection states of DNA. In the majority of bacteriophages studied so far[21,22] the head-to-tail interface is composed of four proteins: the PrtP, two rings of the HCP and at least one TCP. The distal HCP of the *Bacillus subtilis* SPP1 siphovirus was shown to act as the DNA stopper, blocking the phage neck channel by a protruding β-sheet. This element rotates to open a tunnel for DNA release and then closes back up[36]. This DNA gatekeeper function was suggested also for the gp15 bacteriophage T4 HCP[37]. The neck of DT57C differs by two major features. First, instead of comprising four protein rings, DT57C has only three. Second, we found that the neck tunnel at the HCP-TCP interface is open in the pre-ejection DNA state and remains open after DNA

release. In the pre-ejection state, additional density is present in the upper part of the neck tunnel, which contacts the TMP N-terminal moiety (Fig. 2). The HCP of DT57C contains similar β-sheet hairpins (Fig. 2) as the gatekeeper protein of SPP1[36]. Our molecular dynamics analysis suggests that these hairpins are flexible enough to partially occlude the channel (Fig. 2d). Additionally, the inner surface of the lumen possesses a strong negative charge (Fig. 2e), which may hold the DNA in place until ejection.

The presence of the TMP in the tail tube may also help prevent DNA ejection. Upon opening of the tail tip, both TMP (with the exception of its C-terminus) and DNA are ejected. This is evident from the empty tails in our reconstructions and tomograms of the post-ejection state (Fig. 6, Supplementary Fig. 24). It is currently unclear what drives this process: if it is simply the packing pressure of the DNA, or if a small pulling force on the TMP triggers DNA ejection, which would then be sustained by the packing pressure.

The capsid of T5 and other tailed phages are filled with DNA prior to tail attachment[23,24,38]. We hypothesize that the HCP ring initially adopts a closed conformation and its transition to the open conformation is triggered by the binding of pre-assembled tails through the TCP, which becomes intertwined with the HCP to act as an adapter between the C12 and C3 symmetries observed in the PrtP/HCP and TTP rings, respectively. However, despite such a transition to an open state, our cryo-EM map suggests that the DNA does not penetrate beyond the HCP ring in the pre-ejection state.

We identified the TMP as an α-helical trimer by resolving its secondary structural elements in two key locations: at the head-proximal (N-terminal) end of the TMP, and at the distal (C-terminal) end, which retained a proteolytically cleaved TMP fragment in the empty particles as part of the tail tip complex (Fig. 6, Supplementary Fig. 12b). Thus, the overall TMP organization within the tail tube of the phage DT57C can be described as a three-stranded parallel coiled coil, with α-helical secondary structure at least at its N- and C-termini (in accordance with secondary structure prediction). The asymmetric reconstruction of a tail tube segment also revealed the presence of the TMP inside the tail tube. We could not detect any obvious interactions between the inner wall of the tail tube and the TMP in this asymmetric reconstruction of 40 nm long curved segments, suggesting that the TMP may be held in place through non-periodic interactions with the inner walls of the tail tube in addition to contacts established at its C- and N-terminal sections.

The C-terminal TMP fragment obstructing the tail tip channel must undergo significant conformational changes to be temporarily displaced and allow the passage of the bulk of the TMP followed by

DNA during the ejection process. After the DNA is released, the TTHP and C-terminal TMP fragment return to their original conformation. This is supported by the fact that in the tail tip complex of T5, three copies of the C-terminal fragment of TMP are also present in the receptor-bound state, where the channel for DNA ejection is open[12]. Interestingly, the presence of a trimeric C-terminal fragment of the TMP has also been detected for other siphoviruses, including phages 80α[39] (PDB code 6V8I) and λ[40] (PDB code 8IYK). While multiple sequence alignment did not reveal any conserved motif between the TMP C-terminal fragments of DT57C, λ and 80α, the three of them adopt similar structures consisting of a small, trimeric α-helical coiled-coil (Supplementary Fig. 25). The presence of a common fold suggests a conserved function for this region of the TMP, possibly related to the penetration of the bacterial membrane. The DNA release in the T5 phage has been shown to be bi-phasic[41] and this feature is likely conserved in the closely related phage DT57C. However, no other obvious clamp that may temporarily stop the DNA release was identified in the tail structure. We therefore speculate that the fact that the tail tip opening is initially blocked by the TMP fragment may be related to the mechanism arresting the DNA internalization upon insertion of the left-end -9 kbp fragment in vivo[41,42].

In conclusion, we have reconstructed a *Siphovirus* virion at close to atomic resolution, and present atomic models comprising nearly the entire bacteriophage. All-atom molecular dynamics simulations have enabled us to gain additional insights about biological function from the experimentally determined structures. In spite of limitations, such as the attainable timescale and the lack of a possibility to perform a simulation on the atomic model of the entire phage as a whole, our findings set a solid benchmark for future studies to further explore mechanistic and functional aspects of the process of infection by T5-like phages.

## Methods

### Phage generation and purification

Bacteriophages were propagated in a culture of *Escherichia coli* 4 s sensitive strain[13]. The overnight culture was used to inoculate 300 ml of LB medium in 1 L Erlenmeyer flasks, incubated at 37 °C under constant agitation (140 rpm) until $OD_{595}$ reached 0.3. Phage suspension was added at MOI - 10 and cultivation continued for 3 h until visible bacterial lysis, then 0.1% (v/v) chloroform was added to complete lysis. The raw lysate was cleared by high-speed centrifugation (Beckman JA-10 rotor, $15000 \times g$ for 15 min at 4 °C), filter sterilized (0.22 μm PES syringe filter) and pelleted by ultracentrifugation (Beckman 45Ti angle rotor, $75000 \times g$ for 1 h at 20 °C). Phage pellet was resuspended in SM phage buffer ($MgSO_4$ 8 mM, NaCl 100 mM, Tris-HCl 50 mM, pH 7.5), 300 μL of pelleted phage was loaded on top of sucrose step gradient (20-30-40-50-60% (w/v) sucrose in Tris-HCl 50 mM (pH 7.2), NaCl 50 mM, each step ca. 800 μL) in Beckman 5 mL ultracentrifuge tubes. The gradients were run in Beckman SW50.1 rotor for 1 h at $75000 \times g$. Opalescent phage band corresponding to 40-50% sucrose step boundary was collected after removal of top gradient layers containing contaminations (empty and broken phage particles, membrane residues). After an overnight dialysis in Serva dialysis tubing (cut-off 15 kDa) against 1 L of SM phage buffer (the buffer was changed once after 12 h) at 4 °C under constant agitation, phage preparation was strongly opalescent and contained no visible precipitate. It was further concentrated by pelleting phage on a cushion of 1,1,2-trichloro-1,2,2-trifluoroethane (CFC-113) in Beckman SW50.1 bucket rotor for 1 h at $75000 \times g$. A strongly opalescent but not viscous phage concentrate was collected at the water-organic liquid phase interface and stored at 4 °C. This preparation was used for cryo-EM directly. CFC-113 was used as a liquid dampener (1.56 g/mL density, nontoxic, nearly immiscible with water) preventing phage particles from aggregation and breakage under strong centrifugal force.

### Electron microscopy grid preparation

Cryo-EM samples were prepared by depositing 3 μL of purified phage solution onto Quantifoil R2/1 gold grids previously plasma cleaned with a 20% hydrogen, 80% oxygen plasma using a Gatan Solarus (Gatan Inc, USA) for 30 s. Samples were then blotted at 4 °C and 100% humidity with a FEI Vitrobot Mark IV (waiting time 30 s, blotting time 8 s, blotting force 0) before flash-freezing in liquid ethane-propane (50:50).

### Data collection

A set of 34,852 movies were collected on a Titan Krios cryo electron microscope (Thermo Fisher Scientific (TFS)) operating at an acceleration voltage of 300 kV. Images were recorded semi-automatically with EPU software (TFS) with a Falcon 3EC direct electron detector (TFS) in linear mode at a nominal magnification of 78,000, resulting in a pixel size of 1.4 Å/pixel. A defocus range from −0.7 μm to −2.0 μm was applied. The total dose of 67 e−/Å2 was fractionated evenly over 39 frames.

Additionally, 9 tomograms were collected using the same microscope and detector. Tilt series were collected from −60° to 60° with a tilt increment of 2°, target defocus between −4 and −8 μm, a pixel size of 2.2 Å/pixel, and using a dose-symmetric collection scheme[43].

### Image processing

Movies were imported into cryoSPARC and subjected to patch motion correction followed by patch CTF correction. Tail segment particles were picked using cryoSPARC's filament tracer in template-free mode with minimum and maximum filament diameters of 80 Å and 100 Å respectively, and a separation between segments of 1.5 diameters. A total of 7.5 million particles were extracted with a box size of 240 × 240 pixels. The particles were then 2D-classified to remove false positives and noisy particles. The remaining good particles were then separated into two groups, corresponding to tail segments from phage capsids filled with DNA (full) and without DNA (empty). The particles in each group were further selected based on being assigned to 2D classes corresponding to straight tail segments, and were then used to generate initial ab initio 3D models.

To obtain the tail reconstructions, the tail segments of each group were subjected to a heterogeneous 3D refinement with four classes, using as starting models the previously generated ab initio models. Then, the particles corresponding to the class reaching the highest resolution in each case were used to perform a 3D refinement with C3 symmetry against the corresponding model. The resulting maps were inspected to measure helical parameters (rise 41.84 Å, twist 39.51°) for each of the reconstructions. After performing local motion correction and CTF refinement, these helical parameters were imposed in a helical refinement with C3 symmetry, for each of the two sets of particles, with auto tuning of helical parameters enabled.

For reconstructions of the core part of the tail tip, the following WALC (Walking ALignment and Classification) procedure was applied separately to the full and empty segments. Iterative cycles of helical reconstruction, shifting of the resulting map along the Z-axis and re-extraction of particles with the shift in 3D coordinates were performed, together with 2D classification after each re-extraction. Particles showing features corresponding to the tail tip were selected from each 2D classification round. A total of ten iterations with both positive and negative Z-axis shifts were performed, each iteration resulting in a shift of 160 Å along the tail. After pooling the tail tip complex particles obtained from each WALC round (without combining particles corresponding to full and empty phages), duplicates were removed with a proximity threshold of 300 Å. The resulting sets of particles were re-extracted with a box size of 400 × 400 pixels and used to generate ab initio 3D models. The full and empty sets of particles were then refined against the corresponding ab initio model while applying C3 symmetry. Particles were then subjected to local motion correction

and local CTF refinement. For the empty particles, the resulting polished particles were used to perform a final local 3D refinement with C3 symmetry and a mask excluding the pb3 rings. For the full particles, a heterogeneous 3D refinement with three classes was performed. The particles corresponding to the class that reached the highest resolution were selected for final local 3D refinement with C3 symmetry and a mask excluding the pb3 rings. Further local focused refinements were performed using masks covering either the LtfC or pb3 areas, and the resulting maps were combined with phenix.combine_focused_maps to obtain a composite map of the core of the tail tip region.

To obtain reconstructions of the distal region of the tail tip, the particles used for the reconstructions of the core region of the tail tip (both for full and empty states) were shifted by 250 Å along the Z-axis towards the distal end of the tail and re-extracted with a box size of 400 × 400 pixels. The re-extracted particles were then used to generate 3D reconstructions of the distal end without alignment, using the poses from the refinements that yielded the reconstructions of the core tail tip. The resulting reconstructions were used as starting models to perform a final round of local 3D refinement.

Separately, capsid particles were picked using cryoSPARC's template-free blob picker. A total of 5 million particles were extracted with a box size of 900 × 900 pixels. After performing 2D-classification to remove false positives and noisy particles, the remaining good particles were separated into full and empty capsids and used to generate ab initio 3D models.

The capsids of each group were then refined against the corresponding ab initio 3D model with either C5 or icosahedral symmetry. After performing local motion correction and CTF refinement followed by an additional round of refinement, the reconstructions obtained with icosahedral symmetry were used to build an atomic model of the capsid. The C5 reconstructions displayed weak density for the tail connected to the capsid at both of the two vertices aligned with the symmetry axis. To extract particles of the neck region of the phage, multiple rounds of 3D classification into four classes were performed with a mask covering the emerging tail density at either of the two vertices. In each round, particles corresponding to the classes with tail density were selected and carried forward to the next round, until all classes showed strong tail density. Particle coordinates were then re-centered at the coordinates of the corresponding vertex and re-extracted with a box size of 300 × 300 pixels. Both sets of neck region particles (corresponding to full and empty phages) were then merged with particles showing features of the neck region obtained during the previously described WALC procedure, and duplicate particles were removed using a proximity threshold of 400 Å.

The neck particles in each of the two groups were then refined against the model of the capsid with an attached tail obtained during 3D classifications using C6 symmetry. A focused mask excluding the capsid region and the tail tube was created and used in another round of refinement with local angular searches and C12 symmetry. Particles were then subjected to local motion correction and CTF refinement and used to perform a final round of refinement with the same mask and C12 symmetry.

The C12-symmetrized map was subjected to symmetry relaxation to reveal the C6 and C3 parts of the density. We were unable to carry out this procedure in a typical way because the particles were aligning to the strong C12 density of portal and head completion proteins, leaving the C6 and C3 parts of the density smeared. To solve this issue the binned particles aligned to the C12 map were exported to RELION and the symmetry was relaxed as described below. First, the Class3D procedure into four classes with C12 symmetry and no angular searches let us separate the subset of particles that were correctly aligned to the C12 axis from damaged and misaligned particles. Then, the masked Class 3D job was run with the mask covering only the TTP rings. This job used local searches, C3 symmetry and symmetry relaxation option set to C4, which allowed to get C3-symmetrized reference density for the pb6 part of the volume. This reference density and the subset of good particles aligned to C12 symmetry were used in one round of Class3D with two classes, C3 symmetry, C4 symmetry relaxation and local searches with 0.5 degrees range. This allowed us to correctly align particles to the C3 axis and get a low-resolution reference with C12, C6 and C3 regions resolved. Finally, the Refine3D job with the mask excluding the capsid density produced the map with 5.6 Å resolution which served as reference for the final refinements. A further focused refinement from the C3-symmetrized map was performed by local non-uniform refinement using a cylindrical mask including only the tail tube inner channel, to obtain an improved reconstruction of the TMP N-terminus.

For reconstructions of asymmetric, bent segments of the tail tube, tail segments were picked without template from motion-corrected frame sums in cisTEM[44]. After initial 2D classification to remove non-tail images, the remaining 431,751 particles were aligned by global search against a 3D reference consisting of an artificially generated bent feature-less cylinder, without applying any symmetry. The resulting initial reconstruction was then iteratively refined by 3D local angular and shifts refinement. Finally, the data set was split into three 3D classes, separating filled and empty tails, as well as a class containing junk particles. Each class was then further refined by including CTF refinement.

Example micrographs and 2D classes can be found in Supplementary Fig. 26. A summary of the processing workflow is presented in Supplementary Fig. 27. Local resolution maps and FSC curves for all maps are shown in Supplementary Fig. 28.

For tomograms, all movies of each tilt series were first subjected to motion correction with MotionCor2[45]. Then, tilt series were aligned with AreTomo[46] using a binning factor of 4. Finally, after performing initial CTF estimation with CTFPLOTTER[47], 3D CTF correction was applied to the reconstructed tomograms with novaCTF[48]. The number of rings comprising the tail tube of DT57C was determined from the tomograms by measuring the length of the tail from portal of the capsid to the beginning of the tail tip complex, then subtracting the length from the portal to the first TTP ring (determined accurately from the reconstruction of the neck) and dividing by the height of a single TTP ring.

## Model building

Initial models for the tail tip and tail proteins were obtained with Alphafold2[49] or ModelAngelo[50]. The models were filtered by IDDT with phenix.process_predicted_model and fitted into the corresponding density maps. The fitted models were manually inspected and corrected for missing loops and secondary structure elements. The missing parts located in the low SNR map regions were manually modeled with polyalanine to trace the backbone. The map was boxed around the traced models. The boxed maps along with Alphafold2 models were used as an input for phenix.dock_and_rebuild. This approach allowed accurate docking of initial models while avoiding symmetry conflicts and possible model misplacements. Initial models for the LtfA N-terminal parts were built manually with Coot using the sidechain densities as guides for sequence alignment.

Each of the protein models was individually refined against the tail tip density map with phenix.real_space_refine, except for LtfC and LtfA which were refined as a C3 asymmetric subunit containing four copies of LtfC and three copies of LtfA, and pb4, for which only the fragment containing residues 1-325. The refined models were assembled into one file and the inter-chain secondary structure restraints were manually checked with phenix GUI tool. The focused maps along with the half-maps were combined with phenix.combine_maps, using the model file as a reference.

Finally, phenix.real_space_refine was performed with secondary structure, Ramachandran restraints and NCS constraints. Several rounds of manual fixing with ISOLDE[51] were performed. The validation statistics are presented in Supplementary Table 1. Regions of each structural protein present in the built models are presented in Supplementary Table 2.

### Molecular dynamics simulations

The three systems for molecular dynamics simulations consisted of the TTMP tail protein ring along with the adjacent outermost LtfA-LtfC ring (1) and the HCP ring with (2) or without (3) TCP. The protein complexes were placed in a rectangular simulation box (with dimensions of $181 \times 179 \times 81\,\text{Å}/148 \times 148 \times 94\,\text{Å}/148 \times 148 \times 130\,\text{Å}$, respectively) such that the minimal distance between the periodic images was not shorter than 2.4 nm (i.e., twice the cut-off for van der Waals interactions) and solvated with water. The $Na^+/Cl^-$ ions were added such that 0.15 M ionic strength was achieved under the condition of total electroneutrality (total of 368/236, 233/185, and 364/256 Na + /Cl- ions added).

The starting models obtained in this way (containing a total of 255,193/200,302/278,225 atoms, respectively) were subjected to the standard CHARMM-GUI[52] minimization and equilibration protocol as follows. Steepest gradient descent minimization (5,000 steps, maximum force field cut-off 1,000 kJ/mol/nm, Supplementary Fig. 29) was followed by a series of short equilibration simulations (up to 1 ns), with a set of two first equilibration simulations in the canonical NVT (constant volume and temperature) ensemble followed by four equilibration simulations in isothermal-isobaric NPT (constant pressure and temperature) ensemble using the Berendsen thermostat and barostat ($\tau_T = 1.0\,\text{ps}^{-1}$, $\tau_P = 5.0\,\text{ps}^{-1}$) with the harmonic restraints on the protein atoms gradually released. The production simulations were run for 1 μs each. We performed two extra replicate simulations for each production simulation with random velocities drawn from Maxwell's distribution. They resulted in an additional simulation time of 1 μs for the HCP system and 0.66 μs for the HCP + TCP and LtfA-LtfC systems. These replicate simulations consistently agreed with the original simulations, as shown in Supplementary Figs. 8 and 30–34. Cα-atoms of the super-helical C-terminal fragments of LtfA (residues 43-52) and the C-terminal region of HCP (residues 47-108) were constrained during the production simulations by means of harmonic potentials (1000 kJ/mol/nm2) to prevent overall drifting of the complexes and optimize the simulation box size as a result. The convergence of the production simulations was justified based on root mean square deviations of the backbone atoms (the mean RMSD values calculated for the last 500 ns of each MD trajectory are 4.3, 2.3, and 2.2 Å for the three simulated systems, respectively, Supplementary Fig. 8).

Temperature and pressure were set to 303.15 K and 1 bar and controlled by means of the V-rescale thermostat and the Parrinello–Rahman barostat with coupling time constants of $1.0\,\text{ps}^{-1}$ and $5\,\text{ps}^{-1}$, respectively. All MD simulations were carried out in GROMACS version 2020.3[53]. A time step of 1 fs was used at the early steps of equilibration, followed by 2 fs for the final steps and for the production simulation. The covalent bonds to hydrogens were constrained using the LINear Constraint Solver (LINCS) algorithm[54]. The Verlet cutoff scheme and particle mesh Ewald (PME) were used to treat nonbonded interactions. The CHARMM36m force field[55] was used for the proteins with the TIP3P water model, as one of the most advanced and widely used force fields for biomolecular simulations presently available[56].

Secondary structure analysis was performed using the DSSP implementation in MDTraj 1.9.7[57]. The ProDy 2.4.0 toolkit was used to perform principal component analysis[58]. The electrostatic potential was calculated using the APBS server[59] with the default settings and an ionic strength of 0.15 M. The HOLE 2.0 program was used to estimate the inner pore diameter of the HCP ring[60].

### Reporting summary

Further information on research design is available in the Nature Portfolio Reporting Summary linked to this article.

## Data availability

The atomic models generated in this study have been deposited in the Protein Data Bank (PDB) under accession codes 8HQZ (baseplate), 8HRG (helical tail tube), 8HRE (straight tail fibers), 8HQO (neck) and 8HO3 (capsid). Electron density maps have been deposited in the Electron Microscopy Data Bank (EMDB) under accession codes 34955 (baseplate), 34972 (helical tail tube), 34973 (curved tail segment), 34968 (straight tail fibers), 34952 (neck) and 34920 (capsid). Tomograms have been deposited in the EMDB under accession codes 37518, 37519, 37521, 37531, 37534, 37536, 37539, 37543 and 37544. Initial structures for molecular dynamics simulations along with the resulting trajectories have been deposited in Zenodo with identifier 8343960. The structures of the C-terminal fragment of the TMP of other phages are available at the PDB under accession codes 6V8I (80α) and 8IYK (λ). Source data are provided with this paper.

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

## Acknowledgements
This work was supported by the Japanese Ministry of Agriculture and Fisheries (MAFF) (grant GD0696J004 to M.W. and R.A.), the Russian Science Foundation (grant 21-44-07002 to O.S.S.), and the Japan Society for the Promotion of Science (JSPS) (grant 21K20645 to R.A.). Computational simulations were supported by the National Natural Science Foundation of China (grant 32250410316 to P.S.O.).

## Author contributions

A.V.L., O.S.S., and M.W., conceptualization, design of the experiments and supervision. O.S.S. and M.W., funding acquisition. E.E.K., A.K.G., and A.V.L., phage propagation and purification. R.A. and T.-H.C. cryo-EM data collection. R.A., T.-H.C., A.V.M., M.W., and M.A.S. data processing. P.S.O., R.A. and M.A.S., molecular modeling. R.A., A.V.L., T.-H.C., A.V.M., O.S.S. and M.W. writing draft of the manuscript. All authors contributed to expanding and editing the manuscript.

## Competing interests

The authors declare no competing interests.
