## [Peer review file · Nature Communications]

REVIEWER COMMENTS

Reviewer #1 (Remarks to the Author):

The MS "Complete structure of DT57C bacteriophage reveals unusual architecture of head-to-tail interface and lateral tail fibers" describes cryo-EM analysis of the T5-like phage DT57C that is equipped with a set of branched fibers. Ayala et al. purified the phage, imaged it in cryo-electron microscope, performed an extensive image analysis, built atomic models of phage components and performed MD simulations of some of these structures. In summary, this is a pure "structural" paper, where the authors try to derive the function from the structure and come up with various hypotheses and ideas without testing them experimentally.

The impact of the findings is significantly diminished by the availability of two papers that describe the structure and transformation of the T5 tail upon binding to its host cell receptor (PMID 36961893 and 36779755). Those papers are rich in both structural description and functional assays.

In this phage structure expert's opinion, the Ayala et al. paper reports only two findings, not already reported elsewhere for tailed phages in general: 1) the manner by which tail fibers bind to the tail is novel; 2) the presence of a plug, which is interpreted as the C-terminal end of the tape measure protein, in the tip of the tail post DNA release is intriguing. The second point, however, is likely an artifact of sample preparation because a conformation in which the tail is empty, but the tail tip is in a pre-attachment state never occurs in physiological conditions. The tail is emptied when the tail tip binds to BtuB and cocks sideways as it does in T5. The C-terminal end of the TMP will then refold and opens a passage for the TMP. Did the authors consider a possibility that the TMP "floated" inside the capsid in their empty tail particle dataset?

Despite performing a complex and convoluted image reconstruction procedure, the authors appear to not understand the limitations of the resulting datasets. Lines 217-220 read:

"We did not observe any strong interaction interface between the inner lumen of the tail tube and the TMP, indicating that the TMP is held in place mostly through contacts established at its C- and N-terminal sections."

This is incorrect. The contacts could be averaged out because the symmetry of the TMP does not match that of the the TTP. The TMP might be a trimer, but it is not made of disks like the TTP. The averaging procedure (the image reconstruction procedure) is focused on improving the structure of individual TTP subunits, the images are moved and superimposed in a way to maximize that signal. Everything else, like a long filament without periodicity, averages out.

In several instances, the authors use the same approach to interpreting their cryo-EM maps: "if we do not see something, it is not there". But it might not be there because it got averaged out.

The authors constantly refer to a "baseplate" region of the DT57C tail. This phage has no baseplate.

In summary, I find this MS lacking novel functional insights. DT57C is amenable to mutagenesis and the authors can test some of the their ideas with point mutations. The "arms" of the HCP that are proposed to hold the DNA in place could be made "softer" or altered in some other way. Will this lead to premature DNA release or particle instability? Will a non-conserved point mutation in the C-terminal domain of TMP alter the infection properties of the phage?

There are many other comments and concerns listed below with line numbers.

145 The next layer is composed by a hexameric ring of TCP, which is then followed by the first trimeric ring of TCP (Figure 2). ^[1]_{SEP} The acronym TCP must be spelled out here as this is its first mention in the text. The second instance in the sentence is actually TTP, the tail tube protein. It also must be spelled out.

182 To check this hypothesis, we performed a molecular dynamics simulation of the HCP ring alone, which revealed that the β -hairpins of the DI of HCP are prone to large-scale fluctuations resulting in conformations partially occluding the central pore (Figure 2d, Supplementary Movie 1). ^[1]_{SEP} 1) All MD work must be accompanied by plots showing the relaxation of the structure. 2) A few flopping hairpins are proposed to hold 10 atm pressure of packaged DNA. Is this correct? ^[1]_{SEP}

187 This is in accordance with the fact that packed heads must be in a closed state to prevent ejection of the DNA.

The packaged heads must be stable enough for a length of time required to attach the tail during phage assembly inside the host cell.

188 The potential has a strong negative peak at the level of the HCP-TCP interface.

The Methods section states that the e-potential was calculated using APBS? I have calculated the potential of the HCP-TCP ring using the authors' supplied coordinates (the authors' openness in this aspect is much appreciated) in APBS on two different OSs and different versions of APBS and the potential looks "normal", pretty neutral. Some negative and positive patches. Please explain and/or revise. Make sure that you allocate sufficient memory for the APBS program. The HCP-TCP ring required 5.5 GB of RAM with high water requirements of 11 GB.

197 The DNA density is immediately followed by additional density, which we assigned to the TMP N-terminus (Figure 2b). None of the cryo-EM maps contain sufficient information to support this assignment. Please clarify that this assignment is speculative and is based on some other data.

200 Furthermore, our cryo-EM map resolved the secondary structure of the N-terminal part of the TMP, a trimeric α -helical coiled-coil. Sorry, I do not see this in the cryo-EM maps (lowpass filtering the maps in different ways did not help).

209 in a helical symmetry with rise 41.84 Å and twist -39.51. This cannot be right. The twist angle must be positive as the helix is right-handed. I understand that the hand assignment of a helix made of disks is not unique. But one can clearly see in the figures that by rotating one TTP disk clockwise by about 30-60 degrees and moving it up one layer, this disk superimposes onto the next disk. If the rotation was counterclockwise, then it would be -30-60 degrees. I also did the superposition computationally using the authors' provided coordinates and it is about 40 degrees, not -40.

211 Ig-like decoration domain (Figure 5). 1. Why is Fig. 5 mentioned before Fig. 3?

2. None of the maps supplied by the authors contains a density for the Ig domain that would allow interpretation in terms of amino acid positions. In other words, the resolution of the IG domain region in all maps shared by the authors is too low to build an atomic model. If the model of the IG domain is an AlphaFold copy-paste (no rebuilding or it was refined only as a rigid body), it cannot be deposited to PDB as an experimental (experiment-derived) structure. Please remove it from the atomic structure to be deposited to PDB. The domain can certainly be discussed here and it can be shown in figures, but the origin of its "structure" must be clarified in the text and in the figures.

217 We did not observe any strong interaction interface between the inner lumen of the tail tube and the TMP, indicating that the TMP is held in place mostly through contacts established at its C- and N-terminal sections. See my comments above.

223 proteins roughly organized in 3 layers: LtfC, LtfA, BMP, distal tail protein (Dit) and baseplate middle protein (BHP). What LtfC and LtfA stand for? Then, BMP? The last in the list must be the baseplate hub protein (BHP). I do not think it is appropriate to use the word "baseplate" in this paper as this phage does not possess a bona fide baseplate.

277 Finally, DIII comprises two fibronectin type III subdomains, which serve as connection sites for the CFP protein. CFP is mentioned the first time here. We do not know what it is.

290 In order to obtain a reconstruction of the CFP region, we re-extracted particles after applying a downwards shift and performed a refinement with local angular searches. A downward shift"... All particles in the micrographs were oriented with the head pointing up and tail down. And all photographs were always oriented vertically. Simply remove this remark.

298 The cavity enclosed by the BHP also contained additional density for a small α -helical coiled coil

region (Figure 3), which we assigned to be the C-terminus part of the TMP based on its location within the lumen of the tail tube tip and the fact that this region of the TMP is indeed predicted to adopt an α -helical secondary structure. This is not good enough. The TMP must be threaded through the density and the quality of the fit of the atomic model to the density must be evaluated. Ideally, in both orientations because you might be looking at the N-terminus.

To make things easier, I suggest running AlphaFold on small segments of TMP trimers (trimeric state is important) - the size should be equal to the fragment that is resolved in the map - and then use these models to interpret the density. Most likely, AF will predict the structure of the C-terminus correctly.

Otherwise, the interpretation as written represents a search under the lamppost.

312 significant differences between the two states, other than the expected absence of DNA and TMP from the lumen of the neck and tube (Figure 6a). Has the structure of the HCP hairpins, which presumably hold the DNA inside the capsid, changed?

318 terminal region of the TMP to remain enclosed in the tail tip surrounded by pb3 after TMP and DNA ejection (Figure 6b). Is it possible that the rest of the TMP migrated inside the capsid?

364 However, if the tail tip map was rendered at a low contour level (Supplementary Figure 16) we observed weak protrusions corresponding to three fibers attached to the baseplate. We interpret these densities as LtfB trimers that may be folded back and interacting with the baseplate by their C-terminal ends. I am not sure which feature that is described here is shown in Suppl. Fig. 16. Also, the baseplate is mentioned again.

Supplementary Figure 21. Cryo-EM data processing workflow.
The number of particles could be indicated.

Fig. 1. Label the proteins with their names and locations.

Reviewer #2 (Remarks to the Author):

Title; Complete structure of DT57C bacteriophage reveals unusual architecture of head-to-tail interface and lateral tail fibers

Comments; In my view, the results obtained in this study are worthy for publication. The manuscript needs major essential revision before publication. I would like to overview the revised version of the manuscript. I have the following comments/suggestions for authors to address before final decision on the manuscript.

1. "Steepest gradient descent minimization (5,000 steps)": Generally, a 50,000 step energy minimization is performed. Why authors have reduced the number of steps significantly? Also, mention the maximum force cutoff value for energy minimization.
2. "series of short equilibration simulations (up to 1 ns) in the NVT (NPT) ensemble": It is not clear whether, the authors have performed both NVT or NPT simulation. Why NPT is within brackets? Also, authors should define NVT and NPT ensembles.
3. "backbone deviations did not exceed 5 Å throughout the simulation": Mention the average RMSD values.
4. Authors should mention the use of MD simulations in the Abstract at an appropriate place.
5. Clearly define the aim and objectives of the study in the last paragraph of the Introduction section. Discuss the limitations of the study in the end of Discussions section.
6. In the Introduction section the author should refer to the research paper and comment on recent in-silico techniques. It will be good information for the readers. I would like to recommend several papers, among many others, providing further explanation on this topic: PMID: 31903852 PMID: 35362492 PMID: 35276295 PMID: 33465692 PMID: 31138032 PMID: 36925262

7. Authors have not justified the basis of simulation box dimensions; how did they set the box size?
8. In the methodology section number of Na⁺/Cl⁻ ions should be added.
9. The minimization step 5000 is very small. Are the selected systems fully minimized?
10. Authors have to justify the selection of the force field. How it is correct that there are many force fields? Why and on what basis the authors have selected the CHARMM36 force field for simulation?
11. Authors have written, "The temperature and pressure were set to 303.15 K". Even the MD simulations are poorly drafted, and there is no groundwork before the data collection is clearly visible. For example, the human body temperature is 310K, but the author performed at room temperature 303.15 K. While the author is performing only in silico work, why it has not been considered?
12. Rewrite the sentence correctly "series of short equilibration simulations (up to 1 ns) in the NVT (NPT) ensemble using the Berendsen thermostat (and barostat) with the harmonic restraints on protein atoms gradually released."
13. Authors need to elaborate on the data on each secondary structure content.
14. Authors have set the electrostatic potential value as -10 to +10 kT/e. Justify the rationale behind setting the parameter.
15. "binding randomly in any of the a priori equivalent 6 binding modes," Does not make sense.
16. "or straight helical tails, miissing regions" Misspelled word in the line.
17. "Additionally, the inner surface of the lumen possesses a strong negative charge (Figure 2e), which may hold the DNA in place until ejection." Do the authors think the statement is correct? As DNA itself has a negative charge. How a negative charged environment holds a negatively charged DNA molecule.
18. "simulation of the BMP tail protein ring along with the adjacent LtfA-LtfC ring." Do distal tail protein (Dit) and baseplate middle protein (BHP) don't have a significant role in tail ring formation? If they have a role then why do authors exclude them from MD simulations?
19. "LtfC, LtfA, BMP, distal tail protein (Dit), baseplate middle protein (BHP)" Typos error. As the mentioned names for BHP do not match with the names provided in Figure 3. Expanded form for BMP is also missing in the line.

Reviewer #3 (Remarks to the Author):

The authors report the structure of bacteriophage DT57C and complement their structural work with molecular dynamics simulations. The work is interesting and will be of significance to the field. DT57C is related to T5; the authors state that only a low-resolution structure of T5 is available. In an addendum they mention new data reporting on a high-resolution structure of the tail of T5.

The paper is generally straightforward to read and the methodology is well detailed.

Major comments

1. The authors state in a few places that they present the complete/entire/whole structure of DT57C, but this is a bit of an exaggeration. What they mean is that they have determined structures of different parts of the phage e.g. capsid head, tail tube, tail tip, but there are a number of areas where domains were not resolved sufficiently to build an atomic model. In general, more caution should be taken with interpretation throughout the manuscript, especially considering the symmetry that has been imposed. In structure papers it is also useful to report precisely which parts of the proteins are missing (in terms of aa residues).
2. On looking at the supplied structural data, there are some queries. For example, the tail fibre model doesn't seem to fit very well into the map (I'm not including the parts of the protein which are clearly not visible in the maps here). The authors should check and clarify this.
3. Line 120, how is a reasonable threshold defined?
4. Lines 129-138, the electrostatic potential at the center of the hexons part would be easier to follow if the side chains were shown e.g. in a panel in SF3. I find this paragraph a bit speculative, especially as the DCP is not well resolved. Suggest on Line 123, it would be better to say "it is likely that a single

decoration protein copy binds..”

5. Line 149, did the authors try to resolve the DNA e.g. through masking?
6. Line 181, point out the flexible β -hairpins in relevant panels in the figure, e.g. with arrows or colour; I couldn't see them.
7. Lines 193-196, is this data shown?
8. Line 207, it's quite hard to be completely convinced that there are 40 stacked rings. Is there any other evidence for this stoichiometry? The more accurate way to count would be to collect a tomogram of the tail tube.
9. Lines 254-260, can the authors add more detail to explain the significance of the finding that one LftA trimer unfastens?
10. Line 323, point out the “clearly resolved triple-stranded parallel α -helical coiled coil motif” in Fig. 6b and S22f; I couldn't see it.
11. Line 402, it states that TMP is ejected upon opening of the tail tip. I don't follow how this is still present in post-injection state in Fig. 6b. In addition, this figure needs better labelling e.g. what the two colours represent.
12. Line 569, how was the curvature of the feature-less cylinder determined?

Minor comments

1. Line 95, refers to SF1a, which mentions DCP. This has not been introduced yet and I had to read ahead to find out what it referred to.
2. Line 95, “Our high-resolution map led to an atomic model with improved geometry.” Improved compared to what?
3. Line 164, it says 158 in SI
4. Line 211, Figs not in sequence; Fig. 5 referred to before Fig.3
5. Line 328, typo in “missing”
6. Line 832. It would be useful to name the 11 gene products and their colours in the legend.

RESPONSE TO REVIEWER COMMENTS

Original reviewer comments are listed below. Author's responses are in blue and changes referred to by line number in this document are highlighted in the marked-up version of the revised manuscript.

We thank all reviewers for their time and effort to review our manuscript and provide detailed feedback. Thanks to their input, it has become stronger and more concise.

Reviewer #1

The MS "Complete structure of DT57C bacteriophage reveals unusual architecture of head-to-tail interface and lateral tail fibers" describes cryo-EM analysis of the T5-like phage DT57C that is equipped with a set of branched fibers. Ayala et al. purified the phage, imaged it in cryo-electron microscope, performed an extensive image analysis, built atomic models of phage components and performed MD simulations of some of these structures. In summary, this is a pure "structural" paper, where the authors try to derive the function from the structure and come up with various hypotheses and ideas without testing them experimentally.

The impact of the findings is significantly diminished by the availability of two papers that describe the structure and transformation of the T5 tail upon binding to its host cell receptor (PMID 36961893 and 36779755). Those papers are rich in both structural description and functional assays.

We believe that our work provides considerable insights which differ and complement those presented in the cited papers. As the reviewer acknowledges below, we determined how tail fibers bind to the tail and extend from it, displaying a novel structural arrangement not previously observed. Additionally, we reveal the unexpected presence of the C-terminal fragment of the tape measure protein in the post-ejection state (additional data supporting this observation have been added). Furthermore, we have determined the structure of the neck area, not previously described for T5-like phages. This includes the presence of a well-ordered N-terminal fragment of the TMP with a trimeric coiled-coil alpha-helix arrangement, for which we now include improved maps. The atomic models presented here encompass for the first time nearly the complete phage particle. We have expanded the discussion to emphasize on functional aspects that go beyond mere description of structural data.

In this phage structure expert's opinion, the Ayala et al. paper reports only two findings, not already reported elsewhere for tailed phages in general: 1) the manner by which tail fibers bind to the tail is novel; 2) the presence of a plug, which is interpreted as the C-terminal end of the tape measure protein, in the tip of the tail post DNA release is intriguing.

In addition to the mentioned novel findings, our work also describes the architecture of the neck area and the structure of its different components, which has not been previously reported for T5-like phages at this level of detail. Our structure reports a nearly complete atomic structure of all the core components of the DT57C phage for the first time.

The second point, however, is likely an artifact of sample preparation because a conformation in which the tail is empty, but the tail tip is in a pre-attachment state never occurs in physiological conditions. The tail is emptied when the tail tip binds to BtuB and cocks sideways as it does in T5. The C-terminal end of

the TMP will then refold and opens a passage for the TMP. Did the authors consider a possibility that the TMP “floated” inside the capsid in their empty tail particle dataset?

Thank you for this comment. We can now exclude that the presence of the C-terminal fragment of the TMP in the tail tip post DNA release is an artifact. We have performed cryo-electron tomography of DT57C in three different states: empty virus with intact capsid (where the DNA has been ejected), full virus with an intact capsid (where DNA has not been ejected) and empty virus with a broken capsid (see figure below). The tomograms revealed that nearly all empty virions have an intact capsid. Furthermore, empty virions with intact capsids did not contain any TMP density inside the tail tube lumen.

The left panel shows a tomogram of an empty virus with an intact capsid that has released DNA. The capsid is empty, and the tail (red arrow) is also empty. This is evident from the features of the tail, which displays two parallel layers of density, separated by an inner region devoid of density. **The right panel** displays a full virus which has not released DNA. The capsid is filled, and the tail is also filled. This observation is consistent with DNA in the neck region and TMP in the tail region, as described in our single particle reconstructions. Finally, **the middle panel** shows a virus with a broken capsid (yellow arrow). It can be observed that the capsid is empty, due to the release of DNA through the breakage point. However, the tail displays the same features as in the case of the full virus, indicating that it is still filled with the TMP. This indicates that the TMP cannot float back from the tail into empty capsids, as the reviewer has suggested. We have modified the manuscript accordingly to present these new data (lines 312-322)

The tomographs have been deposited to the EMDB database and this figure has been added to the supplementary materials (**Supplementary Figure 24**)

Despite performing a complex and convoluted image reconstruction procedure, the authors appear to not understand the limitations of the resulting datasets. Lines 217-220 read:

“We did not observe any strong interaction interface between the inner lumen of the tail tube and the TMP, indicating that the TMP is held in place mostly through contacts established at its C- and N-terminal sections.”

This is incorrect. The contacts could be averaged out because the symmetry of the TMP does not match that of the the TTP. The TMP might be a trimer, but it is not made of disks like the TTP. The averaging procedure (the image reconstruction procedure) is focused on improving the structure of individual TTP subunits, the images are moved and superimposed in a way to maximize that signal. Everything else, like a long filament without periodicity, averages out.

Thank you for pointing this out. We have now toned down and modified our statement accordingly (lines 208-211).

We would like to highlight that we performed two independent reconstructions of the tail tube: a reconstruction with helical symmetry from segments that appeared locally linear in projection, but also an asymmetric reconstruction of curved tails without imposing any symmetry (**Figure 3**). The latter does not make any prior assumptions about repeating elements. Indeed, although not fully resolved, the tubular TMP density in this asymmetric reconstruction suggests a trimeric coiled-coil arrangement of the TMP., in accordance with the structure observed at the C and N-termini.

In several instances, the authors use the same approach to interpreting their cryo-EM maps: “if we do not see something, it is not there”. But it might not be there because it got averaged out.

Thank you for this comment. We understand these limitations. In this revised version, we checked our manuscript carefully to remove or tone down any such statements that may suggest this interpretation.

The authors constantly refer to a “baseplate” region of the DT57C tail. This phage has no baseplate.

This is another valid point. We have now replaced all occurrences of this incorrect term “baseplate” with “tail tip”.

In summary, I find this MS lacking novel functional insights. DT57C is amenable to mutagenesis and the authors can test some of the their ideas with point mutations. The “arms” of the HCP that are proposed to hold the DNA in place could be made “softer” or altered in some other way. Will this lead to premature DNA release or particle instability? Will a non-conserved point mutation in the C-terminal domain of TMP alter the infection properties of the phage?

As pointed out above, our findings provide substantial novel insights. Taken together with previous studies about the process of assembly of T5-like phages and DNA ejection, our experimentally derived atomic models provide a solid structural basis for mechanistic studies. Numerous novel insights include the structure of the neck region, the structure of the TMP at its N-terminus and C-terminus, the presence of the latter in the post-ejection state, the novel attachment mode of the lateral tail fibers, the MD simulations of the neck-tail adapter, and the first presentation of the nearly-complete structure of a T5-like virus.

There are many other comments and concerns listed below with line numbers.

145 The next layer is composed by a hexameric ring of TCP, which is then followed by the first trimeric ring of TCP (Figure 2). The acronym TCP must be spelled out here as this is its first mention in the text. The second instance in the sentence is actually TTP, the tail tube protein. It also must be spelled out.

Thank you. We have modified the manuscript accordingly (lines 133-137)

182 To check this hypothesis, we performed a molecular dynamics simulation of the HCP ring alone, which revealed that the β -hairpins of the DI of HCP are prone to large-scale fluctuations resulting in conformations partially occluding the central pore (Figure 2d, Supplementary Movie 1).

1) All MD work must be accompanied by plots showing the relaxation of the structure. 2) A few flopping hairpins are proposed to hold 10 atm pressure of packaged DNA. Is this correct?

We have now modified **Supplementary Figure 8** to include all the requested plots, demonstrating that the RMSD values stabilize over the course of the simulations.

Regarding point 2), a detailed inspection of the HCP ring simulation revealed that both observed conformations of the β -hairpins, (1) the “open” pore and (2) the “closed” one are stabilized by specific ionic bridges between highly conserved residues of HCP (K140 with D135 or E145, respectively). Moreover, the addition of the TCP ring seems to further stabilize the “open” conformation due to formation of an alternative ionic contact between E145 of HCP and K41 of TCP. The corresponding figures showing these interactions have been added to the Supplementary Materials (**Supplementary Figures 9-11, Supplementary Movie 2**). The presence of these relatively strong interactions stabilizing the “flopped” β -hairpins (which should be able to tolerate forces of about 1-100 pN each given the typical energy of an ionic bridge of $\sim 1-10$ kcal/mol over the distances of 0.1-1 nm) implies that they can indeed prevent the premature DNA release to a reasonable extent (10 atm pressure corresponds to the force of $1013250 \text{ Pa} \times \pi \times 10^{-18} \text{ m}^2 = \sim 3 \text{ pN}$ when applied to a circular surface with the diameter of 2 nm roughly corresponding to the inner HCP pore). Given that more than one β -hairpin can accommodate the “closed” conformation, several β -hairpins may represent an even stronger barrier for DNA.

This point is now briefly discussed on lines 183-189.

187 This is in accordance with the fact that packed heads must be in a closed state to prevent ejection of the DNA. The packaged heads must be stable enough for a length of time required to attach the tail during phage assembly inside the host cell.

DNA-packed heads have been previously described to be stable, as demonstrated by the fact that they can be purified. This is shown for example in <https://journals.asm.org/doi/10.1128/jvi.02262-13> (mutant T5D17am34d).

188 The potential has a strong negative peak at the level of the HCP-TCP interface.

The Methods section states that the e-potential was calculated using APBS? I have calculated the potential of the HCP-TCP ring using the authors’ supplied coordinates (the authors’ openness in this aspect is much appreciated) in APBS on two different OSs and different versions of APBS and the potential looks “normal”, pretty neutral. Some negative and positive patches. Please explain and/or revise. Make sure that you allocate sufficient memory for the APBS program. The HCP-TCP ring required 5.5 GB of RAM with high water requirements of 11 GB.

We thank the reviewer for pointing to this issue. We have checked the default APBS input parameters used for calculation of the electrostatic potential around the portal complex and found out that they were missing the ionic strength. Therefore, in the revised version we provide more realistic results corresponding to 0.15 M ionic strength. Although it led to significantly lower estimates for the electrostatic potential, the electrostatic profile along the central pore axis still features a negative minimum (i.e., ~ -2.5 kT) at the level of HCP due to the presence of several rings of negatively charged residues, namely D134, E135, and E145. We believe that this region of high negative potential still represents one of the contributing factors preventing premature DNA release, even if not the major one. We have updated the corresponding figure (**Figure 2**) and sections of Results and Methods (lines 174-179, 637-639).

197 The DNA density is immediately followed by additional density, which we assigned to the TMP N-terminus (Figure 2b). None of the cryo-EM maps contain sufficient information to support this assignment. Please clarify that this assignment is speculative and is based on some other data.

We have now obtained improved maps for this region, which show a trimeric coiled-coil structure. We have updated the coordinates and figures accordingly. A detailed view of the fit of the TMP N-terminus can be seen in **Supplementary Figure 12a**. This is consistent with predictions that most regions of TMP, including its N-terminal part, form a coiled coil structure. As such, the density representing TMP can be clearly distinguished from DNA. The part of the manuscript describing these features has been updated (lines 190-194).

Additionally, see analysis of the TMP sequence below by DeepCoil2, a tool for prediction of coiled coil domains, which is in accordance with our findings about the N-terminus of TMP.

200 Furthermore, our cryo-EM map resolved the secondary structure of the N-terminal part of the TMP, a trimeric α -helical coiled-coil. Sorry, I do not see this in the cryo-EM maps (lowpass filtering the maps in different ways did not help).

We have now obtained an improved map for this region, which clearly shows the TMP as a trimeric coiled coil. We have updated the coordinates and figures accordingly (**Supplementary Figure 12a**; lines 190-194).

209 in a helical symmetry with rise 41.84 Å and twist -39.51. This cannot be right. The twist angle must be positive as the helix is right-handed. I understand that the hand assignment of a helix made of disks is not unique. But one can clearly see in the figures that by rotating one TTP disk clockwise by about 30-60 degrees and moving it up one layer, this disk superimposes onto the next disk. If the rotation was counterclockwise, then it would be -30-60 degrees. I also did the superposition computationally using the authors' provided coordinates and it is about 40 degrees, not -40.

Thank you for pointing out this issue. We have now corrected the values (lines 201-202).

211 Ig-like decoration domain (Figure 5).

1. Why is Fig. 5 mentioned before Fig. 3?

2. None of the maps supplied by the authors contains a density for the Ig domain that would allow interpretation in terms of amino acid positions. In other words, the resolution of the IG domain region in all maps shared by the authors is too low to build an atomic model. If the model of the IG domain is an AlphaFold copy-paste (no rebuilding or it was refined only as a rigid body), it cannot be deposited to PDB as an experimental (experiment- derived) structure. Please remove it from the atomic structure to be deposited to PDB. The domain can certainly be discussed here and it can be shown in figures, but the origin of its "structure" must be clarified in the text and in the figures.

We have now removed the Ig-like domains from the deposited PDB structure and modified the text as indicated.

217 We did not observe any strong interaction interface between the inner lumen of the tail tube and the TMP, indicating that the TMP is held in place mostly through contacts established at its C- and N-terminal sections. See my comments above.

This statement has been modified (lines 208-211).

223 proteins roughly organized in 3 layers: LtfC, LtfA, BMP, distal tail protein (Dit) and baseplate middle protein (BHP). What LtfC and LtfA stand for? Then, BMP? The last in the list must be the baseplate hub protein (BHP). I do not think it is appropriate to use the word "baseplate" in this paper as this phage does not possess a bona fide baseplate.

As previously mentioned, we have now replaced instances of "baseplate" by "tail tip". We have replaced the acronyms accordingly (TTMP and TTHP instead of BMP and BHP) and their definitions upon first use (lines 216-218).

277 Finally, DIII comprises two fibronectin type III subdomains, which serve as connection sites for the CFP protein.

CFP is mentioned the first time here. We do not know what it is.

The definition has been added in this first instance (lines 277-278).

290 In order to obtain a reconstruction of the CFP region, we re-extracted particles after applying a downwards shift and performed a refinement with local angular searches. "A downward shift" ... All

particles in the micrographs were oriented with the head pointing up and tail down. And all photographs were always oriented vertically. Simply remove this remark.

Thank you. The remark has been removed.

298 The cavity enclosed by the BHP also contained additional density for a small α -helical coiled coil region (Figure 3), which we assigned to be the C-terminus part of the TMP based on its location within the lumen of the tail tube tip and the fact that this region of the TMP is indeed predicted to adopt an α -helical secondary structure. This is not good enough. The TMP must be threaded through the density and the quality of the fit of the atomic model to the density must be evaluated. Ideally, in both orientations because you might be looking at the N-terminus. To make things easier, I suggest running AlphaFold on small segments of TMP trimers (trimeric state is important) - the size should be equal to the fragment that is resolved in the map - and then use these models to interpret the density. Most likely, AF will predict the structure of the C-terminus correctly. Otherwise, the interpretation as written represents a search under the lamppost.

Thank you for pointing out this unclear description. The procedure described by the reviewer is indeed what we did. We have now rephrased this part and added a figure to show the fitting of the C-terminal part of TMP within the density (**Supplementary Figure 12b**, and lines 287-290)

312 significant differences between the two states, other than the expected absence of DNA and TMP from the lumen of the neck and tube (Figure 6a). Has the structure of the HCP hairpins, which presumably hold the DNA inside the capsid, changed?

We did not observe a significant difference in the conformation of the HCP hairpins. It should be noted that the phages in the empty state still contain the TCP ring inserted into the HCP ring. This is therefore in accordance with our MD analysis, which indicates that insertion of the TCP ring into the HCP ring stabilizes the HCP hairpins into the conformation leaving the tail tube lumen unoccluded.

318 terminal region of the TMP to remain enclosed in the tail tip surrounded by pb3 after TMP and DNA ejection (Figure 6b). Is it possible that the rest of the TMP migrated inside the capsid?

We have now addressed this possibility through cryo-tomography data as described above. An additional supplementary figure has been added (**Supplementary Figure 24**).

364 However, if the tail tip map was rendered at a low contour level (Supplementary Figure 16) we observed weak protrusions corresponding to three fibers attached to the baseplate, We interpret these densities as LtfB trimers that may be folded back and interacting with the baseplate by their C-terminal ends. I am not sure which feature that is described here is shown in Suppl. Fig. 16. Also, the baseplate is mentioned again.

We have now updated the figure to highlight the described feature (now **Supplementary Figure 20**). We have also changed the term "baseplate" to "tail tip".

Supplementary Figure 21. Cryo-EM data processing workflow. The number of particles could be indicated.

We have modified the figure (now **Supplementary Figure 26**) to include the number of particles.

Fig. 1. Label the proteins with their names and locations.

We have now modified the figure as suggested.

We thank the reviewer very much for all their helpful comments.

Reviewer #2

Title; Complete structure of DT57C bacteriophage reveals unusual architecture of head-to-tail interface and lateral tail fibers

Comments; In my view, the results obtained in this study are worthy for publication. The manuscript needs major essential revision before publication. I would like to overview the revised version of the manuscript. I have the following comments/suggestions for authors to address before final decision on the manuscript.

We would like to thank the reviewer for supporting our findings and for their time to review our manuscript.

1. “Steepest gradient descent minimization (5,000 steps)”: Generally, a 50,000 step energy minimization is performed. Why authors have reduced the number of steps significantly? Also, mention the maximum force cutoff value for energy minimization.

We observed that the energy minimization converged to the force cutoff very quickly for all three simulated systems (see new **Supplementary Figure 28**, which shows the corresponding plots of potential energy). Since equilibrium was reached in less than 3,000 steps, we deemed 5,000 steps as sufficient.

Thank you for pointing out the missing maximum force cutoff parameter used during minimization. We have now added its value ($1,000 \text{ kJ mol}^{-1} \text{ nm}^{-1}$) to the Methods section (line 612).

As described in more detail below (point 9), we followed the general, well-established simulation protocol of the CHARMM-GUI web-server. The actual simulations were run on a local machine. This new information has also been added to the methods (lines 609-624).

2. “series of short equilibration simulations (up to 1 ns) in the NVT (NPT) ensemble”: It is not clear whether, the authors have performed both NVT or NPT simulation. Why NPT is within brackets? Also, authors should define NVT and NPT ensembles.

Thank you for pointing out this ambiguity in the description of the equilibration protocol. The initial two steps of equilibration were conducted in the NVT ensemble, while the following 4 steps were conducted in the NPT ensemble. We have now revised this part and defined NVT/NPT meanings (lines 611-617).

3. “backbone deviations did not exceed 5 Å throughout the simulation”: Mention the average RMSD values.

We now provide the average RMSD values in the main text and also present them in **Supplementary Figure 8**.

4. Authors should mention the use of MD simulations in the Abstract at an appropriate place.

We have modified the abstract as suggested (lines 29-30).

5. Clearly define the aim and objectives of the study in the last paragraph of the Introduction section. Discuss the limitations of the study in the end of Discussions section.

We have now modified the manuscript accordingly as suggested (lines 79-83 and 414-421).

6. In the Introduction section the author should refer to the research paper and comment on recent in-silico techniques. It will be good information for the readers. I would like to recommend several papers, among many others, providing further explanation on this topic: PMID: 31903852 PMID: 35362492 PMID: 35276295 PMID: 33465692 PMID: 31138032 PMID: 36925262

We have amended the introduction section as suggested and included all recommended references (lines 79-83).

7. Authors have not justified the basis of simulation box dimensions; how did they set the box size?

Since in all of the simulated systems additional restraints were applied to specific regions in order to prevent overall drifting of the complexes (as stated in the Methods section), we could set the box size in accordance with the dimensions of the simulated systems, i.e. such that the minimal distance between the periodic images of the simulated protein complex was not shorter than 2.4 nm (twice the cut-off value for non-bonded interactions).

Thank you for this comment. This information has now been included and the methods section has been revised (lines 603-606).

8. In the methodology section number of Na⁺/Cl⁻ ions should be added.

We have now specified in the Methods section the number of ions added (lines 606-608).

9. The minimization step 5000 is very small. Are the selected systems fully minimized?

In the choice of minimization step number, we followed the well-established protocol of the CHARMM-GUI web-server by Lee et al [10.1021/acs.jctc.5b00935] for biomolecular simulations. Given that according to this protocol, the initial energy minimization is followed by a series of equilibration simulations with backbone harmonic restraints, we concluded that it resulted in sufficiently equilibrated systems within the chosen number of steps. We now provide the corresponding plots in the Supplementary Materials (**Supplementary Figure 28**). The methods section has been updated accordingly (lines 611-613).

10. Authors have to justify the selection of the force field. How it is correct that there are many force fields? Why and on what basis the authors have selected the CHARMM36 force field for simulation?

We only used a single force field. We chose the CHARMM36m force field because it is the most recent version of the popular CHARMM force field family featuring the modified CMAP backbone potential for better simulations of both folded proteins and IDPs [10.1016/j.sbi.2018.02.002],

and it is available in the highly parallelizable and efficient Gromacs MD engine [10.1021/ct900549r].

We have now included a statement about our choice to the revised manuscript. (lines 631-634).

11. Authors have written, "The temperature and pressure were set to 303.15 K". Even the MD simulations are poorly drafted, and there is no groundwork before the data collection is clearly visible. For example, the human body temperature is 310K, but the author performed at room temperature 303.15 K. While the author is performing only in silico work, why it has not been considered?

Since the phage we have studied here infects *E. coli* bacterial cells (which also happens outside the human body), our choice of simulation temperature has been guided by typical experimental lab conditions for growing such phages and bacteria. It has been reported that infectivity of T5- and T4-like phages does not change significantly within the temperature range 22-37C [10.3389/fmicb.2021.616712]. In addition, 303.15 K is a standard temperature used for the parametrization of the CHARMM family of the force fields, which is commonly used – we thus chose this temperature in our simulations to facilitate comparisons with previous research.

12. Rewrite the sentence correctly "series of short equilibration simulations (up to 1 ns) in the NVT (NPT) ensemble using the Berendsen thermostat (and barostat) with the harmonic restraints on protein atoms gradually released."

We thank the reviewer for pointing this out. We have rephrased this sentence as suggested (lines 611-617).

13. Authors need to elaborate on the data on each secondary structure content.

We have now included discussions on relevant secondary structure elements of the different structural components of the phage (lines 180-189, 253-259).

14. Authors have set the electrostatic potential value as -10 to +10 kT/e. Justify the rationale behind setting the parameter.

The range for the electrostatic potential was guided by the minimum and maximum values of the estimated potential. Based on a suggestion by Reviewer #1, we found that we had not included an appropriate ionic strength. The calculation of electrostatic potential was repeated; results and description have been updated (lines 174-179, 637-639).

15. "binding randomly in any of the a priori equivalent 6 binding modes," Does not make sense.

Thanks for pointing this out. Since hexons that compose the capsid have local 6-fold symmetry, it would be expected that a copy of the decoration protein binding to its center would be able to bind in 6 possible equivalent orientations. We have now shortened and modified this section to improve clarity (line 119).

16. "or straight helical tails, miissing regions" Misspelled word in the line.

This has been corrected (line 328).

17. “Additionally, the inner surface of the lumen possesses a strong negative charge (Figure 2e), which may hold the DNA in place until ejection.” Do the authors think the statement is correct? As DNA itself has a negative charge. How a negative charged environment holds a negatively charged DNA molecule.

We propose a possible mechanism for preventing premature release of DNA from filled tail-less capsids based on electrostatic repulsion between the HCP lumen, which has high negative electrostatic potential, along with the partial occlusion of the HCP lumen by the flexible β -hairpins of HCP protein. The negatively-charged environment would be able to hold back DNA since, in order for DNA to be ejected, it would have to overcome the repulsion caused by the region of the inner surface of the lumen with strong negative charge.

18. “simulation of the BMP tail protein ring along with the adjacent LtfA-LtfC ring.” Do distal tail protein (Dit) and baseplate middle protein (BHP) don't have a significant role in tail ring formation? If they have a role then why do authors exclude them from MD simulations?

We have limited the size of the simulated systems in order to allow for a longer simulation time scale.

19. “LtfC, LtfA, BMP, distal tail protein (Dit), baseplate middle protein (BHP)” Typos error. As the mentioned names for BHP do not match with the names provided in Figure 3. Expanded form for BMP is also missing in the line.

We thank the reviewer for pointing out these issues, which have now been corrected (lines 216-218).

Reviewer #3

The authors report the structure of bacteriophage DT57C and complement their structural work with molecular dynamics simulations. The work is interesting and will be of significance to the field. DT57C is related to T5; the authors state that only a low-resolution structure of T5 is available. In an addendum they mention new data reporting on a high-resolution structure of the tail of T5.

The paper is generally straightforward to read and the methodology is well detailed.

We thank the reviewer for their positive comments and their time reviewing our manuscript.

Major comments

1. The authors state in a few places that they present the complete/entire/whole structure of DT57C, but this is a bit of an exaggeration. What they mean is that they have determined structures of different parts of the phage e.g. capsid head, tail tube, tail tip, but there are a number of areas where domains were not resolved sufficiently to build an atomic model. In general, more caution should be taken with interpretation throughout the manuscript, especially considering the symmetry that has been imposed.

In structure papers it is also useful to report precisely which parts of the proteins are missing (in terms of aa residues).

We have updated the text with special attention to the mentioned term (lines 29, 85, 99, 415). Additionally, we have now added a table indicating the parts of each protein that are present (**Supplementary Table 2**).

2. On looking at the supplied structural data, there are some queries. For example, the tail fibre model doesn't seem to fit very well into the map (I'm not including the parts of the protein which are clearly not visible in the maps here). The authors should check and clarify this.

We have now removed the coordinates for the Ig-like domain from the tail PDB, for which only weak density was observed. Additionally, for the CFP region, we have now also deposited an additional map with the unsharpened reconstruction, which better visualizes domains in the most distal and flexible regions.

3. Line 120, how is a reasonable threshold defined?

Here we used a map contour level of 3σ above average, which excluded most of the noise and unconnected blobs. We have now specified the value in the legend of **Supplementary Figure 2**.

4. Lines 129-138, the electrostatic potential at the center of the hexons part would be easier to follow if the side chains were shown e.g. in a panel in SF3. I find this paragraph a bit speculative, especially as the DCP is not well resolved. Suggest on Line 123, it would be better to say "it is likely that a single decoration protein copy binds.."

We have now modified the figure (**Supplementary Figure 3**) and text as suggested (legend of **Supplementary Figure 2**).

5. Line 149, did the authors try to resolve the DNA e.g. through masking?

We thank the reviewer for their suggestion. We have attempted extensive masking as suggested, and while we did not obtain significant improvements for the DNA itself, the strategy was successful at improving the N-terminus of TMP region (**Supplementary Figure 12a**). We have now updated figures and manuscript to include this new result (lines 190-194).

6. Line 181, point out the flexible β -hairpins in relevant panels in the figure, e.g. with arrows or colour; I couldn't see them.

Thank you. We have now modified the figure as suggested (**Figure 2d**).

7. Lines 193-196, is this data shown?

Thank you for pointing this out. We have now added it a proper reference to the display items showing this data (line 183).

8. Line 207, it's quite hard to be completely convinced that there are 40 stacked rings. Is there any other evidence for this stoichiometry? The more accurate way to count would be to collect a tomogram of the tail tube.

We have now collected tomography data of DT57C. We have measured the number of rings of the tail tube from tomograms as suggested, obtaining a matching value of 40 rings per tail, in accordance with the previous value measured from individual particles. An example of a tomogram is presented in **Supplementary Movie 3**. Tomograms have been uploaded to the EMDB (lines 199-201)

9. Lines 254-260, can the authors add more detail to explain the significance of the finding that one LftA trimer unfastens?

We have now extended this paragraph as suggested (lines 245-259).

10. Line 323, point out the “clearly resolved triple-stranded parallel α -helical coiled coil motif” in Fig. 6b and S22f; I couldn't see it.

We have now added an additional figure to show the mentioned region more clearly (**Supplementary Figure 12b**).

11. Line 402, it states that TMP is ejected upon opening of the tail tip. I don't follow how this is still present in post-injection state in Fig. 6b. In addition, this figure needs better labelling e.g. what the two colours represent.

Thank you for pointing out this confusing statement. While it is true that indeed we find the bulk of the TMP to be ejected upon opening of the tail tip, we found its C-terminal fragment to still be present in the post-injection state. This is the density labeled as TMP C-terminus in **Figure 6b**. As discussed at the end of the Results section and later in the Discussion section, this indicates that this fragment is proteolytically cleaved from the rest of the TMP, in accordance with previous observations. Furthermore, it suggests that during the process of ejection of the DNA and the bulk of the TMP, the tail tip complex and the C-terminus of TMP must undergo a significant conformational change to allow the opening of a passage for the DNA and bulk of TMP to be ejected. After ejection is completed, such conformational changes would be reverted, resulting in the observed post-ejection state which retains the C-terminus of TMP.

The explanation has been clarified (lines 303-311, 399-408). We have also added the meaning of the two colors in **Figure 6** as suggested. An additional experiment using electron tomography and corresponding figure about the presence and absence of the TMP has been added (**Supplementary Figure 24**).

12. Line 569, how was the curvature of the feature-less cylinder determined?

We determined it by matching its curvature to the curvature of prominent 2D classes. This artificial alignment reference was only used for the initial alignment round at low resolution (50 Å). All subsequent 3D refinements used the resulting 3D reconstruction from the preceding cycle. A comment has been added (lines 558-565).

Minor comments

1. Line 95, refers to SF1a, which mentions DCP. This has not been introduced yet and I had to read ahead to find out what it referred to.

We have now defined DCP and MCP before referring to **Supplementary Figure 1a** (line 104).

2. Line 95, “Our high-resolution map led to an atomic model with improved geometry.” Improved compared to what?

We meant improved geometry compared to the previous model available for the capsid of T5 (<https://www.rcsb.org/structure/6omc>).

3. Line 164, it says 158 in SI

This has now been corrected (line 152).

4. Line 211, Figs not in sequence; Fig. 5 referred to before Fig.3

We have now corrected the numbering of the Figures.

5. Line 328, typo in “missing”

This has now been corrected (line 328).

6. Line 832. It would be useful to name the 11 gene products and their colours in the legend.

We have now labeled the location of each protein within the context of the full virus in panel **b**, and as previously mentioned we have added a table indicating the parts of each protein that are present (**Supplementary Table 2**)

REVIEWER COMMENTS

Reviewer #1 (Remarks to the Author):

"Nearly complete structure of bacteriophage DT57C reveals unusual architecture of head-to-tail interface and lateral tail fibers" is an improved version of the MS previously titled "Complete structure of DT57C bacteriophage reveals unusual architecture of head-to-tail interface and lateral tail fibers". The authors addressed many points raised by the reviewers, but there are still a few sticking points that the authors may want to consider.

First, the title. The "unusual" architecture of the head-to-tail interface has not been covered in sufficient detail in the MS. To claim an unusual architecture, much more careful analysis is required that should include 1) bioinformatics that shows that the neck of this phage is different from that of other long tailed phages; 2) AlphaFold modeling of said necks for which the structure is unknown; 3) comparison with several recently reported myophage neck structures (PMID: 37422479, 36656854, 37684529). In the opinion of this scientist, the organization of the T5 neck is pretty "standard".

The second concern is that the paper is presented as if the tail structure of the homologous T5 phage tail has not been published and described in great detail (e.g. line 61 in the Intro: However, only a low-resolution structure of T5 is available¹²). DT57C and T5 carry different tail fibers. This MS is an excellent opportunity to compare the structures and to show their common elements and where and how they diverge. Table 1 is too simple.

The third concern is that the authors do not attempt to reconcile their structure of the emptied particle with the structure of the T5 tail bound to FhuA. Is the C-terminal fragment of the TMP present in the FhuA-bound T5 tail (in the cryoEM maps - it might have been omitted from the atomic model)? What is the conformation of the tail in the surface-bound particle? Does it contain the C-terminal fragment of the TMP? Can the C-terminal fragment of the TMP plug the channel tight enough to prevent the leakage of ions or larger molecules?

Lastly, I am still confused by the authors' interpretation of the cryoEM density of the TMP in the tail's lumen. The TMP appears to be trimeric at its both extremes. The authors interpret the diffuse density in the curved mid-section reconstruction of the tail as the "structure of the TMP":

L. 914-920 "The asymmetric reconstruction clearly revealed the presence of the TMP inside the tail tube. Along its extension through the tail tube, the TMP adopts a hollow elongated barrel-like conformation, resembling a curved helical tube. We could not detect any obvious interactions between the inner wall of the tail tube and the TMP in this asymmetric reconstruction of 40 nm long curved segments, suggesting that the TMP may be held in place through non-periodic interactions with the inner walls of the tail tube in addition to contacts established at its C- and N-terminal sections." First of all, the interpretation of cryoEM density, which is a hypothesis in this case, should be given in the Discussion part of the MS instead of the figure legend. Second, this "hollow tube" structure is clearly incompatible with the trimeric structure of the TMP termini and with the secondary structure prediction (or AlphaFold prediction), which shows that TMPs in long tailed phages are long alpha-helices. To my knowledge, the first reliable demonstration of the structure of the TMP in the lumen of a long-tailed phage was presented in Fig. 4 of this paper (which, unfortunately, is not without its own major drawbacks). Note, that in that myophage structure paper, the TMP was visualized because the reconstructed volume did not "slide" along the length of the tube, as is the case for the asymmetric reconstruction of the DT57C tube presented here. My point: the structure of the TMP cannot be derived from the cryoEM density presented in Fig. 3d. What we see here is an artifact of image reconstruction procedure.

A few additional line-by-line comments are below.

L. 86. "base plate"

L. 87-88. "instead of four". N.B. T4 is special as it carries a tail tube terminator gp3. Most long tailed

phages do not have a special tail tube terminator. Only tail (sheath, if present) terminator.

L. 133-134. "The portal protein (PrTP) forms a dodecameric ring, which is inserted directly into the capsid in place of one of the MCP pentons." I do understand who and why "inserted" the portal into the capsid. The portal protein replaces one of the vertex pentamers in the capsid shell. Also of note, phage people, in general, do not like calling capsid pentamers and hexamers "pentons" and "hexons" leaving those terms to eukaryotic virus people that study jelly roll-based capsid protomers.

L. 166-167. "the density attributed to the DNA disappeared abruptly (Figure 2b)." I actually do not see an "abrupt disappearance" of the density. In fact, any truly sharp feature in a cryoEM or X-ray map is a result of map manipulation - merging, masking, etc., which clearly took place here. I encourage the authors to do a better job at masking and deciding (guessing, more like it) where DNA ends and where TMP starts because the DNA-TMP segmentations shown in Fig. 2a and 2b are different.

L. 176-177. "The negative potential peaks at the level of the HCP-TCP interface." Does "peaks" mean that the potential has a maximum at that position?

L. 180. Is the described observation really "remarkable"?

L. 210-211. "Within the lumen of the entire tail, TMP adopts an elongated hollow tubular conformation." "Hollow tubular conformation" - I do not know what this is, and how one can derived this from the presented data.

L. 259. "whole LftA ring" - entire LftA ring.

L. 280 "a shift away from the core of the tail tip". I do not understand.

Going back to the structure of the TMP.

1. In all figures showing the atomic structure of the TMP (Fig. 2a, Fig 4, Fig. 6, Supplementary Fig. 12), show residue numbers. Referring to the model as N- and C-terminal part is not just imprecise, it is very annoying. I have no idea which parts of the chain were modeled after scouting the MS text and figures several times.

2. Show a fragment of the electron density with fitted side chains. In Supplementary Fig. 12, I see some random helices fitted into the electron density. This figure carries zero information for me.

3. Show AlphaFold prediction (with predicted accuracy) next to the model refined against the cryoEM map.

4. It would be good to compare the structure of the TMP (the C-term of the TMP) to several structures already available. Maybe some interesting and universal properties of TMPs can be derived from this analysis?

Reviewer #2 (Remarks to the Author):

The authors have responded to all concerns meticulously and improved the manuscript accordingly. The revised draft is improved significantly. I do not have further comments. I recommend the revised draft for publication.

Reviewer #3 (Remarks to the Author):

The authors have satisfactorily addressed all of my comments and improved the manuscript.

We thank the reviewer for their time and thorough evaluation of our manuscript. We have addressed all raised concerns in the response below.

Reviewer #1 (Remarks to the Author): “Nearly complete structure of bacteriophage DT57C reveals unusual architecture of head-to-tail interface and lateral tail fibers” is an improved version of the MS previously titled “Complete structure of DT57C bacteriophage reveals unusual architecture of head-to-tail interface and lateral tail fibers”. The authors addressed many points raised by the reviewers, but there are still a few sticking points that the authors may want to consider.

First, the title. The “unusual” architecture of the head-to-tail interface has not been covered in sufficient detail in the MS. To claim an unusual architecture, much more careful analysis is required that should include 1) bioinformatics that shows that the neck of this phage is different from that of other long tailed phages; 2) AlphaFold modeling of said necks for which the structure is unknown; 3) comparison with several recently reported myophage neck structures (PMID: 37422479, 36656854, 37684529). In the opinion of this scientist, the organization of the T5 neck is pretty “standard”.

We have now removed “unusual” from the title and any occurrences in the text.

The second concern is that the paper is presented as if the tail structure of the homologous T5 phage tail has not been published and described in great detail (e.g. line 61 in the Intro: However, only a low-resolution structure of T5 is available¹²). DT57C and T5 carry different tail fibers. This MS is an excellent opportunity to compare the structures and to show their common elements and where and how they diverge. Table 1 is too simple.

We have now reworded the mentioned part of the introduction (lines 62-65). Furthermore, we have extended Table 1 to provide more detailed information about the degrees of identity and similarity between equivalent proteins in DT57C and T5.

The third concern is that the authors do not attempt to reconcile their structure of the emptied particle with the structure of the T5 tail bound to FhuA. Is the C-terminal fragment of the TMP present in the FhuA-bound T5 tail (in the cryoEM maps - it might have been omitted from the atomic model)? What is the conformation of the tail in the surface-bound particle? Does it contain the C-terminal fragment of the TMP? Can the C-terminal fragment of the TMP plug the channel tight enough to prevent the leakage of ions or larger molecules?

We have now incorporated the findings of the FhuA-bound state including citation and reconciled these with our structures in an updated Discussion (lines 425-427). Inspection of our tail tip reconstruction reveals that it is completely sealed.

Lastly, I am still confused by the authors’ interpretation of the cryoEM density of the TMP in the tail’s lumen. The TMP appears to be trimeric at its both extremes. The authors interpret the diffuse density in the curved mid-section reconstruction of the tail as the “structure of the TMP”: L. 914-920 “The asymmetric reconstruction clearly

revealed the presence of the TMP inside the tail tube. Along its extension through the tail tube, the TMP adopts a hollow elongated barrel-like conformation, resembling a curved helical tube. We could not detect any obvious interactions between the inner wall of the tail tube and the TMP in this asymmetric reconstruction of 40 nm long curved segments, suggesting that the TMP may be held in place through non-periodic interactions with the inner walls of the tail tube in addition to contacts established at its C- and N-terminal sections.” First of all, the interpretation of cryoEM density, which is a hypothesis in this case, should be given in the Discussion part of the MS instead of the figure legend. Second, this “hollow tube” structure is clearly incompatible with the trimeric structure of the TMP termini and with the secondary structure prediction (or AlphaFold prediction), which shows that TMPs in long tailed phages are long alpha-helices. To my knowledge, the first reliable demonstration of the structure of the TMP in the lumen of a long-tailed phage was presented in Fig. 4 of this paper (which, unfortunately, is not without its own major drawbacks). Note, that in that myophage structure paper, the TMP was visualized because the reconstructed volume did not “slide” along the length of the tube, as is the case for the asymmetric reconstruction of the DT57C tube presented here. My point: the structure of the TMP cannot be derived from the cryoEM density presented in Fig. 3d. What we see here is an artifact of image reconstruction procedure.

We have now moved the interpretation of the asymmetric tail tube reconstruction into the Discussion section as suggested (lines 416-421). Furthermore, we agree that the averaging of segments located at different positions along the tail tube limits the extent to which the resulting density can be interpreted. We have therefore removed the claim that it adopts a hollow tubular structure.

A few additional line-by-line comments are below.

L. 86. “base plate”

We have now replaced this by “tail tip” (line 90).

L. 87-88. “instead of four”. N.B. T4 is special as it carries a tail tube terminator gp3. Most long tailed phages do not have a special tail tube terminator. Only tail (sheath, if present) terminator.

We have now rephrased this part to take this fact into account (lines 91-92).

L. 133-134. “The portal protein (PrpP) forms a dodecameric ring, which is inserted directly into the capsid in place of one of the MCP pentons.” I do understand who and why “inserted” the portal into the capsid. The portal protein replaces one of the vertex pentamers in the capsid shell. Also of note, phage people, in general, do not like calling capsid pentamers and hexamers “pentons” and “hexons” leaving those terms to eukaryotic virus people that study jelly roll-based capsid protomers.

We have now modified the wording of this part as suggested (lines 137-138). Additionally, we have replaced the occurrences of “pentons” and “hexons” by “pentamers” and “hexamers” respectively, as suggested.

L. 166-167. “the density attributed to the DNA disappeared abruptly (Figure 2b).” I actually do not see an “abrupt disappearance” of the density. In fact, any truly sharp feature in a cryoEM or X-ray map is a result of map manipulation - merging, masking, etc., which clearly took place here. I encourage the authors to do a better job at masking and deciding (guessing, more like it) where DNA ends and where TMP starts because the DNA-TMP segmentations shown in Fig. 2a and 2b are different.

We have identified the previous sharp feature mentioned by the reviewer, which was due to an error during the process of making the figures. We have now modified the figure with a correct, updated version
Additionally, we have modified the wording “abrupt disappearance” to describe more accurately what we observe (line 170), which is remarkable reduction in the diameter of the observed density (but not a sharp disappearance). We have assigned the density up to the HCP included to be DNA based on this fact, plus the fact that it is connected with the DNA density contained in the capsid (but not with the density that we assign to be TMP).

L. 176-177. “The negative potential peaks at the level of the HCP-TCP interface.” Does “peaks” mean that the potential has a maximum at that position?

We have now rephrased this statement, stating the observation that the value of the negative potential reaches a minimum at that level (lines 180-181).

L. 180. Is the described observation really “remarkable”?

This adjective has been removed (line 185).

L. 210-211. “Within the lumen of the entire tail, TMP adopts an elongated hollow tubular conformation.” “Hollow tubular conformation” - I do not know what this is, and how one can derived this from the presented data.

As mentioned in a previous comment, we have now moved and modified the interpretations related to this reconstruction to the Discussion section (lines 416-421). We have removed the quoted statement from the Results section.

L. 259. “whole LftA ring” - entire LftA ring.

We have now modified the wording as suggested (lines 263-264).

L. 280 “a shift away from the core of the tail tip”. I do not understand.

We have now reworded this statement to make it clearer (lines 286-288).

Going back to the structure of the TMP.

1. In all figures showing the atomic structure of the TMP (Fig. 2a, Fig 4, Fig. 6, Supplementary Fig. 12), show residue numbers. Referring to the model as N- and C-

terminal part is not just imprecise, it is very annoying. I have no idea which parts of the chain were modeled after scouting the MS text and figures several times.

The information has now been added to the relevant figures as suggested.

2. Show a fragment of the electron density with fitted side chains. In Supplementary Fig. 12, I see some random helices fitted into the electron density. This figure carries zero information for me.

*We have now added the suggested panel to **Supplementary Figure 12**, together with information about the N and C-terminal parts that were built as previously indicated.*

3. Show Alphafold prediction (with predicted accuracy) next to the model refined against the cryoEM map.

*We have now added the suggested panels to **Supplementary Figure 12**.*

4. It would be good to compare the structure of the TMP (the C-term of the TMP) to several structures already available. Maybe some interesting and universal properties of TMPs can be derived from this analysis?

*We have identified the structures of the C-terminus of TMP of phages 80 α and lambda (unpublished, PDB code 8IYK) as available. As suggested, we have now included a figure showing multiple sequence alignment and structural comparison of the C-terminal TMP fragments of DT57C, 80 α and lambda (**Supplementary Figure 25**).*

REVIEWERS' COMMENTS

Reviewer #1 (Remarks to the Author):

The authors have addressed most of my comments. However, a few small things remain. I suggest leaving these to authors' discretion.

Considering that phage proteins have a modular organization, Table 1 could have an additional column called "Matching regions", which specifies the boundaries of domains for which the identity/similarity is given in the neighboring column.

A correction to my previous comments: "...To my knowledge, the first reliable demonstration of the structure of the TMP in the lumen of a long-tailed phage was presented in Fig. 4 of this paper (which, unfortunately, is not without its own major drawbacks)." The PubMed ID of the paper in question was mentioned earlier, but got deleted here for some reason: 36656854. Tubes of density in that map could be fitted with extended alpha-helices and were interpreted as the TMP.

I am surprised that there is some sort of a resistance about adding residue number to the actual ribbon diagrams. At least the N- and C-terminal residues should be labeled with the actual number next to the structure or with a line pointing to that point on the structure and residue number. For example, I do not know where residues 1192 and 1227 are in Suppl. Fig. 25a. Is the structure oriented with the N-terminus pointing up? I picked Suppl. Fig. 25a because I am looking at it now. But this applies to ALL ribbon diagram figures. N.B. There are no residue numbers in Suppl. Fig. 14c. and 17b where they would be quite useful.

This is probably too late at this point, as all the figures are done. I strongly advice against using shadows in ribbon diagrams. They might look cool at first sight, but they bring an extra level of complexity instead of clarity. Consider that without shadows there would be fewer lines to figure out in Suppl. Fig. 14a (in all ribbon diagrams).

Suppl. Fig. 12. A repeat of the above question/concern: is it possible to label the N- and C-terminal residues in the actual panels?

Suppl. Fig. 12c. Are residue numbers correct? Perhaps, this region can be indicated with a box in panel a? This looks like a C-terminal fragment to me, so residue numbers would be incorrect. This region could be indicated in panel b with a box or with two flanking residue numbers if the box is too intrusive.

Suppl. Fig. 12d, 12e. No residue numbers, again. I won't comment on other occurrences, but, in general, this aspect of the paper is still not great.

Fig. 2e. Is there a reason for choosing such a narrow range of e-potential color code ($\pm 3kT/e$)? Both Chimera and ChimeraX can color the clipped surface any solid color (e.g. white). This will make the panel on the right less confusing.

We would like to thank reviewer #1 for their dedication in improving our manuscript.

Reviewer #1 (Remarks to the Author):

The authors have addressed most of my comments. However, a few small things remain. I suggest leaving these to authors' discretion.

Considering that phage proteins have a modular organization, Table 1 could have an additional column called "Matching regions", which specifies the boundaries of domains for which the identity/similarity is given in the neighboring column.

Thank you for this suggestion. We have now added the suggested column.

A correction to my previous comments: "...To my knowledge, the first reliable demonstration of the structure of the TMP in the lumen of a long-tailed phage was presented in Fig. 4 of this paper (which, unfortunately, is not without its own major drawbacks)." The PubMed ID of the paper in question was mentioned earlier, but got deleted here for some reason: 36656854. Tubes of density in that map could be fitted with extended alpha-helices and were interpreted as the TMP.

I am surprised that there is some sort of a resistance about adding residue number to the actual ribbon diagrams. At least the N- and C-terminal residues should be labeled with the actual number next to the structure or with a line pointing to that point on the structure and residue number. For example, I do not know where residues 1192 and 1227 are in Suppl. Fig. 25a. Is the structure oriented with the N-terminus pointing up? I picked Suppl. Fig. 25a because I am looking at it now. But this applies to ALL ribbon diagram figures. N.B. There are no residue numbers in Suppl. Fig. 14c. and 17b where they would be quite useful.

As suggested, we have now added residue numbers to all figures that did not have them (Supplementary Figures 12, 14, 17 and 25).

This is probably too late at this point, as all the figures are done. I strongly advice against using shadows in ribbon diagrams. They might look cool at first sight, but they bring an extra level of complexity instead of clarity. Consider that without shadows there would be fewer lines to figure out in Suppl. Fig. 14a (in all ribbon diagrams).

We thank the reviewer for this suggestion. We will consider it for future work.

Suppl. Fig. 12. A repeat of the above question/concern: is it possible to label the N- and C-terminal residues in the actual panels?

We have now labeled these panels as suggested.

Suppl. Fig. 12c. Are residue numbers correct? Perhaps, this region can be indicated with a box in panel a? This looks like a C-terminal fragment to me, so residue numbers

would be incorrect. This region could be indicated in panel b with a box or with two flanking residue numbers if the box is too intrusive.

We thank the reviewer for pointing this out. We have now corrected these residue numbers and have highlighted the region in panel b of the same figure with a box, as suggested.

Suppl. Fig. 12d, 12e. No residue numbers, again. I won't comment on other occurrences, but, in general, this aspect of the paper is still not great.

These panels and others throughout the Figures have now been labeled.

Fig. 2e. Is there a reason for choosing such a narrow range of e-potential color code ($\pm 3kT/e$)? Both Chimera and ChimeraX can color the clipped surface any solid color (e.g. white). This will make the panel on the right less confusing.

The range of potential values was chosen to match the values in the potential plot presented in the left panel of Figure 2e. We have now updated the right panel of Figure 2e to color the clipped surface in a different color (light grey) as suggested.